# Conservation of preparatory neural events in monkey motor cortex regardless of how movement is initiated

**Antonio H Lara[1], Gamaleldin F Elsayed[1,2], Andrew J Zimnik[1], John P Cunningham[2,3,4], Mark M Churchland[1,3,5,6]\***

[1]Department of Neuroscience, Columbia University Medical Center, New York, United States; [2]Center for Theoretical Neuroscience, Columbia University, New York, United States; [3]Grossman Center for the Statistics of Mind, Columbia University Medical Center, New York, Unitedstate; [4]Department of Statistics, Columbia University, New York, United States; [5]David Mahoney Center for Brain and Behavior Research, Columbia University Medical Center, New York, United States; [6]Kavli Institute for Brain Science, Columbia University Medical Center, New York, United States

**Abstract** A time-consuming preparatory stage is hypothesized to precede voluntary movement. A putative neural substrate of motor preparation occurs when a delay separates instruction and execution cues. When readiness is sustained during the delay, sustained neural activity is observed in motor and premotor areas. Yet whether delay-period activity reflects an essential preparatory stage is controversial. In particular, it has remained ambiguous whether delay-period-like activity appears before non-delayed movements. To overcome that ambiguity, we leveraged a recently developed analysis method that parses population responses into putatively preparatory and movement-related components. We examined cortical responses when reaches were initiated after an imposed delay, at a self-chosen time, or reactively with low latency and no delay. Putatively preparatory events were conserved across all contexts. Our findings support the hypothesis that an appropriate preparatory state is consistently achieved before movement onset. However, our results reveal that this process can consume surprisingly little time.
DOI: https://doi.org/10.7554/eLife.31826.001

**\*For correspondence:**
mc3502@columbia.edu

**Competing interests:** The authors declare that no competing interests exist.

## Introduction

Multiple lines of evidence suggest that voluntary movement is preceded by a preparatory stage (*Tanji and Evarts, 1976*; *Rosenbaum, 1980*; *Wise, 1985*; *Riehle and Requin, 1993*; *Ghez et al., 1997*; *Crammond and Kalaska, 2000*; *Rickert et al., 2009*; *Michaels et al., 2015*; *Bastian et al., 2003*; *Rosenbaum, 2010*). Most fundamentally, the voluntary reaction time (RT) typically becomes shorter when a delay period separates an instruction from a go cue, presumably because the delay allows preparation time to complete (*Rosenbaum, 1980*; *Riehle and Requin, 1993*; *Michaels et al., 2015*; *Churchland et al., 2006*). Neurons in many brain areas — including primary motor cortex (M1) and dorsal premotor cortex (PMd) — respond selectively during the delay (*Tanji and Evarts, 1976*; *Riehle and Requin, 1993*; *Weinrich et al., 1984*; *Godschalk et al., 1985*; *Wise and Kurata, 1989*; *Kurata, 1989*; *Crammond and Kalaska, 1989*). Delay-period activity is thus hypothesized to reflect a preparatory process, albeit one that becomes artificially sustained due to the imposed delay. In support of this interpretation, delay-period activity is predictive of RT variability (*Riehle and Requin, 1993*; *Michaels et al., 2015*; *Bastian et al., 2003*; *Churchland et al., 2006*; *Afshar et al., 2011*) and its disruption erases RT-savings provided by the delay (*Churchland and Shenoy, 2007a*).

Although appealing, aspects of this interpretation remain incomplete, uncertain, or controversial. There exist at least three alternative interpretations where delay-period activity is either not preparatory or not essential. First, delay-period activity may be primarily (*Wong et al., 2015*) or partially (*Perfiliev et al., 2010*; *Wise et al., 1986*) suppressive, reflecting the need to inhibit movement until the go cue. Second, delay-period activity may be facilitatory but inessential and may thus be bypassed when there is no delay (*Ames et al., 2014*). Third, delay-period activity may reflect a higher-level 'planning' process essential only when movements are complex (*Wong et al., 2015*) or involve a non-direct motor mapping (*Perfiliev et al., 2010*) (e.g. between joystick and cursor motion). In support of the latter proposal, direct reaches can be executed at such short latencies that 'it is not possible to regard the response as voluntarily planned' (*Perfiliev et al., 2010*) under the typical assumption that motor preparation is a time-consuming process (*Rosenbaum, 1980*; *Riehle and Requin, 1993*; *Ghez et al., 1997*; *Bastian et al., 2003*; *Churchland et al., 2006*). Related behavioral results similarly argue that preparation is either unnecessary or much faster than typically assumed (*Wong et al., 2015*; *Haith et al., 2016*). Behind the above alternatives lies a deeper concern: perhaps delay-period activity is specific to an artificial situation where an experimenter-imposed delay interrupts the normal pacing of voluntary movement.

We test the hypothesis that delay-period activity reflects an essential preparatory stage yet can occur unexpectedly rapidly. A central prediction of that hypothesis is that activity resembling delay-period activity should still occur, briefly, when the delay duration is zero. This has been explored by multiple studies but results have remained ambiguous. An early study revealed that some aspects of delay-period activity are recapitulated without a delay (*Crammond and Kalaska, 2000*), as did (*Churchland et al., 2006*). However, using state-space methods, (*Ames et al., 2014*) found that the delay-period neural state was bypassed in zero-delay trials. This result has been variously interpreted as evidence that delay-period activity is beneficial but inessential (*Ames et al., 2014*), that delay-period activity is suppressive (*Wong et al., 2015*), or that the range of acceptable preparatory states is broad (*Michaels et al., 2015*). Yet other findings of Ames et al. suggested that some early, and putatively preparatory, response aspects are present without a delay.

Here we compare neural activity with and without a delay but employ a different analysis approach and consider neural activity across a broader range of behavioral contexts. To observe neural activity during an imposed delay, we used the standard delayed-reach paradigm with an explicit go cue (the 'cue-initiated context'). To examine preparation without a delay, we did not simply reduce the cue-initiated delay to zero (as in *Crammond and Kalaska, 2000*; *Churchland et al., 2006*; *Ames et al., 2014*) but instead adopted the paradigm of (*Perfiliev et al., 2010*) and used moving targets. Because moving targets evoke particularly low-latency reaches, this 'quasi-automatic' context provides a stringent test of the prediction that delay-period-like activity should appear in the absence of a delay. We also employed a 'self-initiated' context: monkeys were free to reach upon target onset but received larger rewards if they chose to wait. This context was designed to distinguish between the competing hypotheses that delay-period activity is preparatory versus suppressive. If preparatory, delay-period-like activity should be present in the self-initiated context and should grow with time. If suppressive, any such activity should wane with time.

As in prior studies, a technical hurdle is determining whether the patterns of neural activity during a delay reappear without a delay. Most neurons with delay-period activity also exhibit movement-related activity, making it challenging to determine whether activity in a narrow time-window is preparation-related, movement-related, or some combination. To overcome this challenge, we leveraged the recent finding (*Elsayed et al., 2016*) that delay-period activity and movement-related activity occupy orthogonal population-level subspaces. Based on the cue-initiated context, we used delay-period responses to define a putatively 'preparatory subspace' and responses during movement to define a 'movement subspace'. We then quantified events in these subspaces during the other contexts.

Preparatory- and movement-subspace activity patterns were surprisingly well-conserved across contexts. For self-initiated reaches, preparatory subspace activity grew with time, becoming strongest before movement onset. This finding agrees with the hypothesis that such activity plays a preparatory rather than suppressive role. Quasi-automatic reaches exhibited the same sequence of preparatory- and movement-subspace events as cue-initiated and self-initiated reaches. However, that sequence unfolded remarkably rapidly: preparatory-subspace events led movement-subspace events by only a few tens of milliseconds. We have hypothesized that a goal of movement

preparation is to initialize movement-generating dynamics (*Churchland et al., 2006*; *Churchland et al., 2010*; *Churchland et al., 2012*; *Shenoy et al., 2013*; *Churchland and Cunningham, 2014*; *Sussillo et al., 2015*; *Michaels et al., 2016*). Consistent with a key prediction of this hypothesis, each reach direction was preceded by a preparatory-subspace state that was similar across contexts. Thus, our results support the hypothesis that delay-period activity reflects a preparatory process that consistently occurs even when there is no delay. However, in agreement with (*Perfiliev et al., 2010*; *Haith et al., 2016*), our results argue against the idea that movement preparation necessarily involves time-consuming cognitive or high-level planning processes. It is more likely that preparatory activity plays a straightforward and mechanistic role: initializing the networks that will produce descending motor commands.

## Results

We trained two monkeys (Ba and Ax) to execute reaches from a central touch-point to radially arranged targets. We employed three contexts. The cue-initiated context (*Figure 1a*) emulated the standard instructed-delay paradigm: a variable delay period (0–1000 ms) separated target onset from an explicit go cue. A key hypothesis being examined is that motor preparation can occur during the delay and is reflected in delay-period neural activity. We thus analyzed trials with delays > 400 ms, allowing observation of sustained delay-period activity. Shorter delays ensured the go cue was unpredictable, encouraging timely and consistent preparation.

In the self-initiated context (*Figure 1b*), monkeys were free to reach upon target presentation but waiting yielded larger rewards up to a limit at 1200 ms. Growing target size mirrored the growing reward. The self-initiated context was included to distinguish between the hypotheses that delay-period activity is primarily preparatory versus suppressive. The preparatory hypothesis predicts that delay-period-like activity should be present in the self-initiated context and should grow stronger in anticipation of execution. Conversely, the suppressive hypothesis predicts that delay-period-like activity should wane as execution becomes more probable and suppression is relieved.

The quasi-automatic context (*Figure 1c*) was similar to the cue-initiated context, but the go cue was the onset of target motion along a radial path toward the screen's edge. Monkeys had to intercept the target mid-flight as it 'escaped'. Initial target position was calibrated such that interception occurred close to the target position in the other contexts. To recapitulate the traditional comparison between trials with zero delay versus a long delay (*Crammond and Kalaska, 2000*; *Churchland et al., 2006*; *Ames et al., 2014*), we analyzed only zero-delay quasi-automatic trials and compared with cue-initiated trials possessing a delay >400 ms (see above). This provides a particularly stringent comparison because moving targets evoke quite low-latency responses (see below). Given this time pressure, delay-period-like events are expected to be observed during the quasi-automatic context only if they are part of the typical process of movement preparation and initiation, and not if they are suppressive or are otherwise an 'artificial' product of the experimenter-imposed delay. The quasi-automatic context also allows exploration of the possibility that preparation is beneficial but 'skipped' when movements are hurried (*Ames et al., 2014*).

Touch-point/target color specified context (*Figure 1*). Trials for the three contexts were randomly interleaved. Monkeys successfully completed most trials: 91% and 94% of cue-initiated trials (monkey Ba and Ax respectively), 92% and 96% of self-initiated trials, and 93% and 93% of quasi-automatic trials. Failures were distributed across a variety of modes, summarized in *Table 1*.

### Reaction times (RTs)

RT was defined as the time between the visual go cue (confirmed by a photodetector) and a change in hand velocity (*Methods*). Cue-initiated RTs were 253 ± 50 ms and 235 ± 37 ms (mean ±standard deviation) for monkey Ba and Ax (*Figure 2a,b*, *red traces*). These RTs are on the brisk side of the range in prior studies, consistent with the goal of the cue-initiated context: to encourage monkeys to prepare during the delay and reach promptly after the go cue. Monkeys were almost always successful in waiting for the go cue; overt reaches during the delay occurred on <1% of trials for both monkeys.

In the self-initiated context, monkeys typically waited at least 600 ms before reaching but rarely waited until the time of maximum reward at 1200 ms. We define RT in the self-initiated context as the interval between target and movement onset. Thus, RT has a unified definition across contexts:

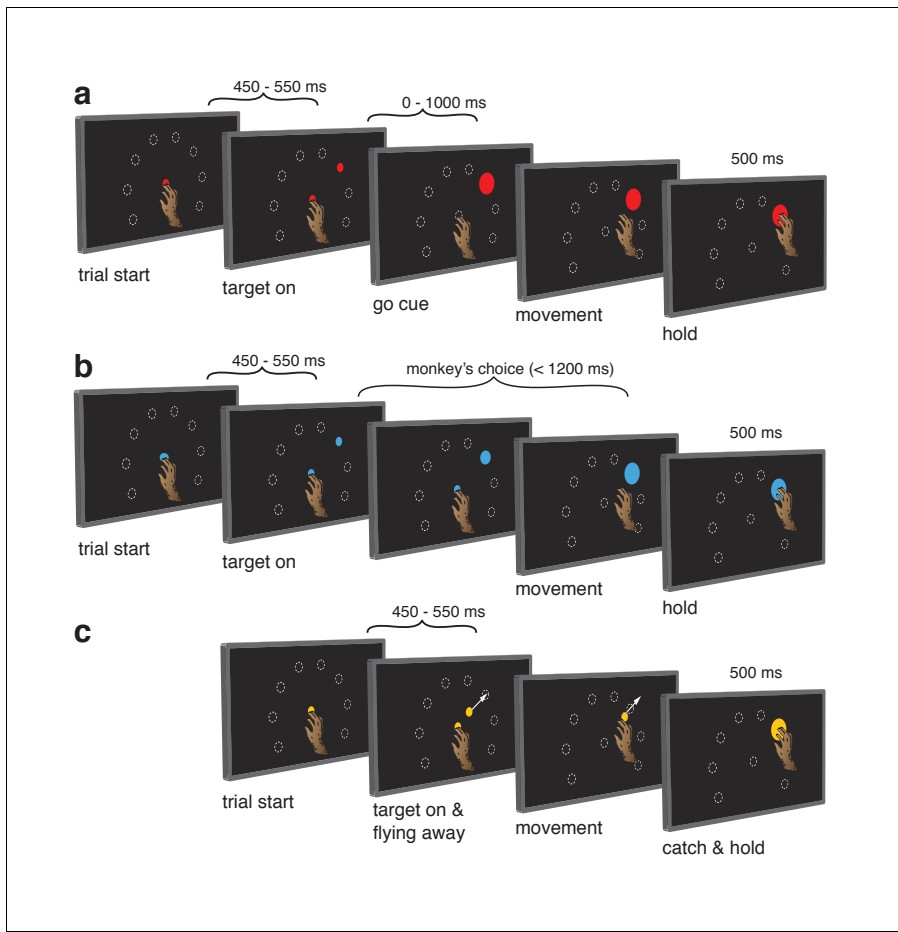

**Figure 1.** Behavioral task. Monkeys performed the same set of reaches under three initiation contexts. (a) Cue-initiated context. Each trial began when the monkey touched a red central point on the screen. After a brief hold period (450–550 ms) a red target appeared in one of eight possible locations (white dashed circles, not visible to the monkey) 130 mm from the touch point. After a variable delay period (0–1000 ms) the target suddenly increased in size providing the go cue to initiate the reach. (b) Self-initiated context. Trials began as above, but the central point was blue. Subsequently, a small blue target appeared and gradually grew in size. Monkeys were free to initiate the reach as soon as the target appeared on the screen. However longer waiting times were rewarded with juice rewards. (c) Quasi-automatic context. The central point was yellow. Yellow targets appeared in one of eight possible locations. The initial target location was 40 mm from the touch point. Immediately after appearing, the target moved radially outward. Monkeys had to intercept the target before it reached the edge of the screen and disappeared. Given typical reaction times, target interception occurred at a location near the location of the targets in the other two tasks (dashed circles).
DOI: https://doi.org/10.7554/eLife.31826.002

the time of movement initiation relative to the time when movement was first permitted. Self-initiated RTs were 935 ms ± 132 ms and 996 ms ± 106 ms for the two monkeys (*Figure 2a,b*, *blue*). The considerable RT variability is unlikely to reflect uncertainty regarding reward size, which was directly conveyed by target size. RT variability presumably reflects the natural tension between a desire for large reward and a desire for immediate reward, with different factors dominating on different trials.

Quasi-automatic RTs were 205 ± 32 ms and 192 ms ± 27 ms for monkey Ba and Ax (*Figure 2a,b*, *yellow*). Despite the lack of time to prepare in advance, quasi-automatic RTs were shorter than cue-initiated RTs: by 48 and 43 ms on average. Yet monkeys almost never 'jumped the gun'; reaches were rare during the 100 ms after target onset (<1% of trials for both monkeys). This was unsurprising: target location was unknown ahead of time, making it impossible to successfully exploit a strategy of anticipatory reaches.

**Table 1.** Rates of different categories of task error.

Percentages consider trials that were successfully initiated (i.e. the monkey held the touch point and a target appeared) but were subsequently failed. The target was missed if a reach was executed but did not land within the acceptance window. Delay violations occurred if above-threshold hand velocity was detected between target onset and the go cue. This could occur if the monkey made an overt reach or a small adjustment. Trials were also failed if hand velocity was above threshold during the interval from the go cue until 100 ms later. Changes in hand velocity during this time are unlikely to reflect legitimate responses to the go cue, and were thus treated the same way as movement during the delay. Movement during the target-hold period (after it was successfully hit) also resulted in a failure. This occurred if velocity exceeded threshold, or if hand position exited the acceptance window. The maximum RT (500 ms for cue-initiated, 1500 ms for self-initiated) elapsed if the hand never moved. Such failures tended to happen if the monkey became distracted during a trial, or near the end of the day as motivation waned. For example, simply holding the hand on the screen but not performing the task resulted in such failures being recorded. Violations of the maximum movement duration occurred if the hand left the touch-point but did not land on the target within 500 ms. This could occur for very sluggish movements, or if the monkey simply aborted the trial and did not attempt to hit the target. We also included in this category failures in the quasi-automatic context where the hand did not land back on the screen before the target disappeared off the screen.

| | Monkey ba | | | Monkey ax | | |
|---|---|---|---|---|---|---|
| | Cue-initiated | Self-initiated | Quasi-automatic | Cue-initiated | Self-initiated | Quasi-automatic |
| Missed target | 1.4% | 4.1% | 3.3% | 0.9% | 1.6% | 2.4% |
| Delay violation (adjustment or overt reach) | 2.2% | NA | NA | 1.3% | NA | NA |
| Velocity violation within 100 ms of go cue (adjustment or non-physiological RT) | 1.6% | 0.1% | 0.6% | 1.1% | 0% | 0.5% |
| Moved during target-hold period | 1.4% | 1.3% | 2.1% | 1.5% | 1.2% | 1.1% |
| Max RT elapsed | 2.2% | 2.6% | NA | 0.8% | 0.6% | NA |
| Max movement duration elapsed | 0.2% | 0.2% | 1.3% (target left screen) | 0.1% | 0.2% | 2.7% (target left screen) |

DOI: https://doi.org/10.7554/eLife.31826.003

## The quasi-automatic context yields low-latency muscle responses

Changes in electromyographic (EMG) activity occurred, as expected, before detectable changes in hand velocity (*Figure 3*, in each panel compare *black trace* with *yellow trace* at top). We assessed the earliest onset of EMG in the quasi-automatic context via two approaches. First, for all muscles, we computed trial-averaged rectified EMG, time-locked to target onset (*Figure 3*, one *colored trace* per condition). The earliest change typically occurred around 100 ms. Second, we examined single-trial EMG latencies for those muscles where baseline activity was low enough to allow EMG onset to be confidently identified on most trials. EMG onset could be rapid: as early as 90 ms after the go cue for monkey Ba (*Figure 4* and *Figure 4—figure supplement 1*) and 110 ms for monkey A (*Figure 4—figure supplement 2*). These low latencies are consistent with the earliest onset of EMG in (*Perfiliev et al., 2010*): 90–110 ms across subjects. Low-latency responses were typical: the majority of trials had EMG latencies shorter than 144 ms (*red line*), the mean latency in (*Perfiliev et al., 2010*). RTs based on hand velocity were slightly longer than in (*Perfiliev et al., 2010*) (where the mean RT ranged from 120 to 200 ms across monkeys and ~139–189 ms across humans). Slightly longer RTs are expected in the present task; velocity cannot rise quickly until the hand is lifted off the screen, resulting in a modestly greater lag (~40–80 ms) between EMG onset and a measured change in velocity.

The low latency of quasi-automatic reaches aids interpretation when asking whether delay-period-like events occur without a delay. This question has traditionally been addressed by including zero-delay trials that are otherwise identical to long-delay trials (*Crammond and Kalaska, 2000*; *Churchland et al., 2006*; *Ames et al., 2014*). While valid, that approach raises a potential concern: might monkeys, once accustomed to a delay, start to treat zero-delay trials as if they have a delay? For example, might monkeys increase their RT to accommodate the expectation of a delay and begin to generate delay-period-like neural events that would otherwise not occur? The low-latency of quasi-automatic trials addresses such concerns. Monkeys cannot be 'adding' their own delay,

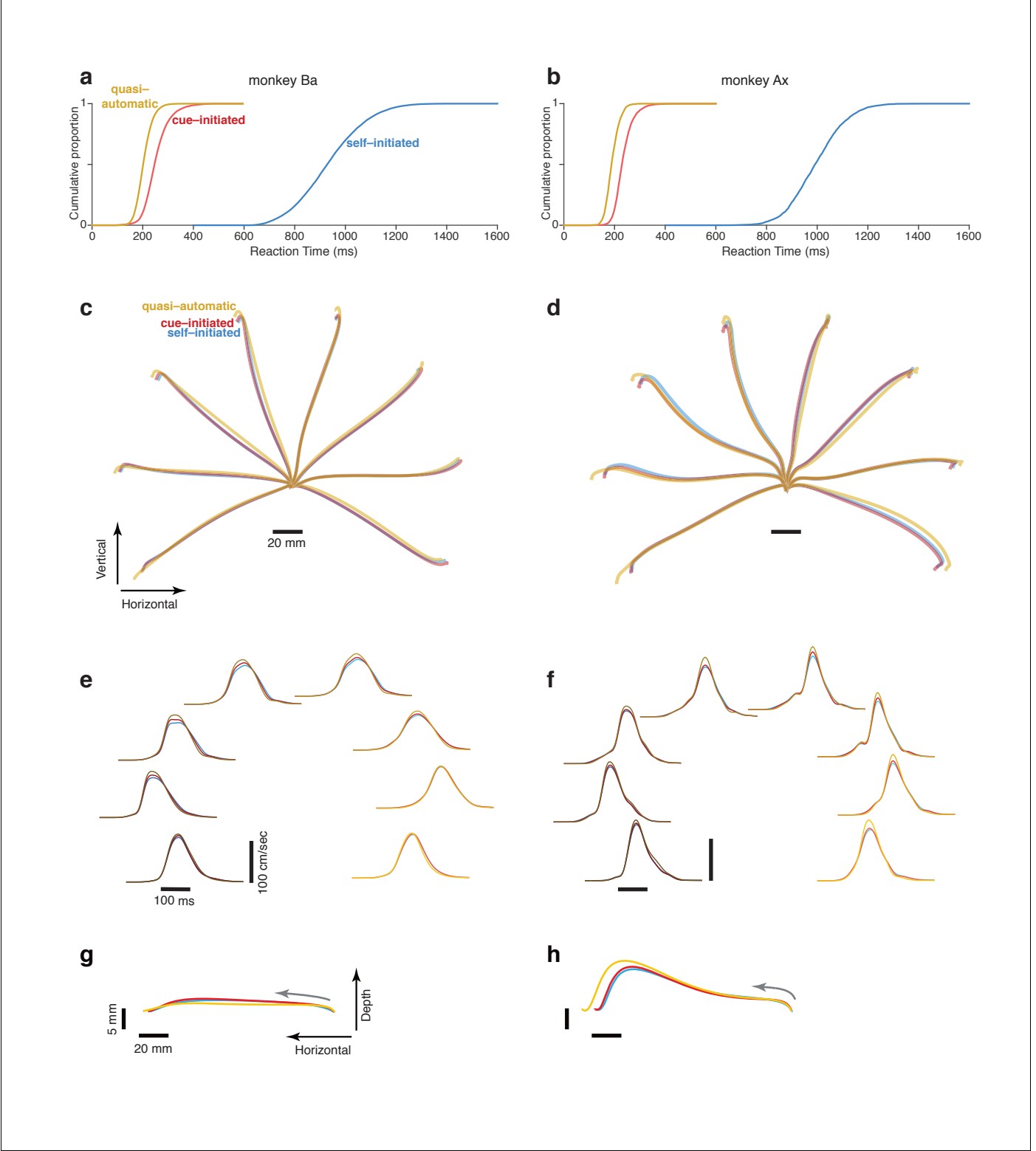

**Figure 2.** Reaction time distributions and reach kinematics. (a,b) cumulative reaction time distributions for the three contexts. Trials are pooled across all recordings. Data for monkey Ba and Ax are shown in the left and right columns respectively. (c,d) Average reach trajectories for the eight targets and three contexts. (e,f) Average speed profiles in the three contexts, using the same color coding as above. Line shade indicates reach direction: *light traces* for rightwards reaches and *dark traces* for leftwards reaches. This same shade-coding is preserved in subsequent figures. (g,h) Average reach trajectories for one example reach direction (leftward) with depth shown on an expanded scale to allow closer examination of trajectories in that dimension. Gray arrows indicate direction in which the hand traveled.

*Figure 2 continued on next page*

*Figure 2 continued*

DOI: https://doi.org/10.7554/eLife.31826.004

The following figure supplements are available for figure 2:

**Figure supplement 1.** Individual-trial reach trajectories from a typical session.

DOI: https://doi.org/10.7554/eLife.31826.005

**Figure supplement 2.** Distributions of initial reach angle during the quasi-automatic context.

DOI: https://doi.org/10.7554/eLife.31826.006

otherwise RTs would be longer rather than shorter than for the cue-initiated context. Indeed, quasi-automatic responses are sufficiently swift that it becomes reasonable to question whether preparation is possible (*Perfiliev et al., 2010*; *Haith et al., 2016*). That said, we do not argue that the quasi-automatic context represents a different 'class' of movement. Rather, we leverage the quasi-automatic context as a stringent test of the hypothesis that delay-period events may occur even without a delay.

## Reach kinematics are well-matched across contexts

The comparison of neural activity across contexts is most straightforward if reach kinematics are matched (otherwise differences might relate to preparation of different reaches, rather than context per se). The match in kinematics was indeed quite good (*Figure 2c–h*, *Figure 2—figure supplement 1*) with only small differences across contexts. Quasi-automatic reaches tended to have a slightly greater extent and a slightly higher peak velocity (on average 7% and 9% higher relative to cue-initiated movements) but were otherwise very similar to cue-initiated and self-initiated reaches. It may seem surprising that quasi-automatic reaches did not display dramatically higher velocities. However, we find that monkeys typically reach rapidly even when not required, presumably out of a desire to obtain reward quickly once the decision to move has been made.

Might monkeys sometimes begin quasi-automatic reaches before they are fully specified (*Ghez et al., 1997*), which might cause preparation to be trivially absent? This was not the case: even the earliest reaches were consistently target directed (*Figure 2—figure supplement 2*) as in (*Perfiliev et al., 2010*), and hand trajectories did not exhibit guesses with subsequent corrections (*Figure 2—figure supplement 1*). Yet while quasi-automatic movements were consistently target-directed, they were slightly less accurate: the circular standard deviation (*Berens, 2009*) of initial reach direction was 17% and 11% higher (monkey Ba and Ax; $p < 0.001$ for both). Furthermore, quasi-automatic trials with RTs shorter than the median were slightly less accurate than those with RTs longer than the median: the standard deviation of reach direction was 4.9% and 3.0% higher ($p < 0.05$ for monkey Ba, NS for monkey Ax). In absolute terms these effects were small – the standard deviation of initial reach direction was never higher than 4.3 degrees. Still, reaches that missed the target were slightly more common in the quasi-automatic context: an increase from 1.4% (cue-initiated) to 3.3% (quasi-automatic) for monkey Ba and from 0.9% to 2.4% for monkey Ax. In summary, reach kinematics were well matched across contexts but reaches were slightly less accurate in the quasi-automatic context, as might be expected given their low latency.

## Muscle activity patterns are well-matched across contexts

As with kinematics, muscle activity (EMG) should ideally be similar across contexts to aid comparisons of neural activity. We computed trial-averaged muscle responses using the same approach used for neural responses (see below). EMG response patterns were very similar across contexts (*Figure 3—figure supplement 1*). During movement, the median correlation between the self-initiated and cue-initiated contexts was 0.98 and 0.98 (monkey Ba and Ax respectively; activity compared in a 300 ms window starting 50 ms before movement onset; median taken across muscles). The median correlation between quasi-automatic and cue-initiated contexts was 0.97 and 0.96. EMG magnitude (quantified as the standard deviation across directions and time) was also similar across contexts, with some small differences. Magnitude was slightly higher for the quasi-automatic context (7.2% and 8.8% for monkey Ba and Ax) relative to the cue-initiated context, consistent with the slightly faster reach kinematics. Magnitude was slightly lower for the self-initiated context (by 1.9%

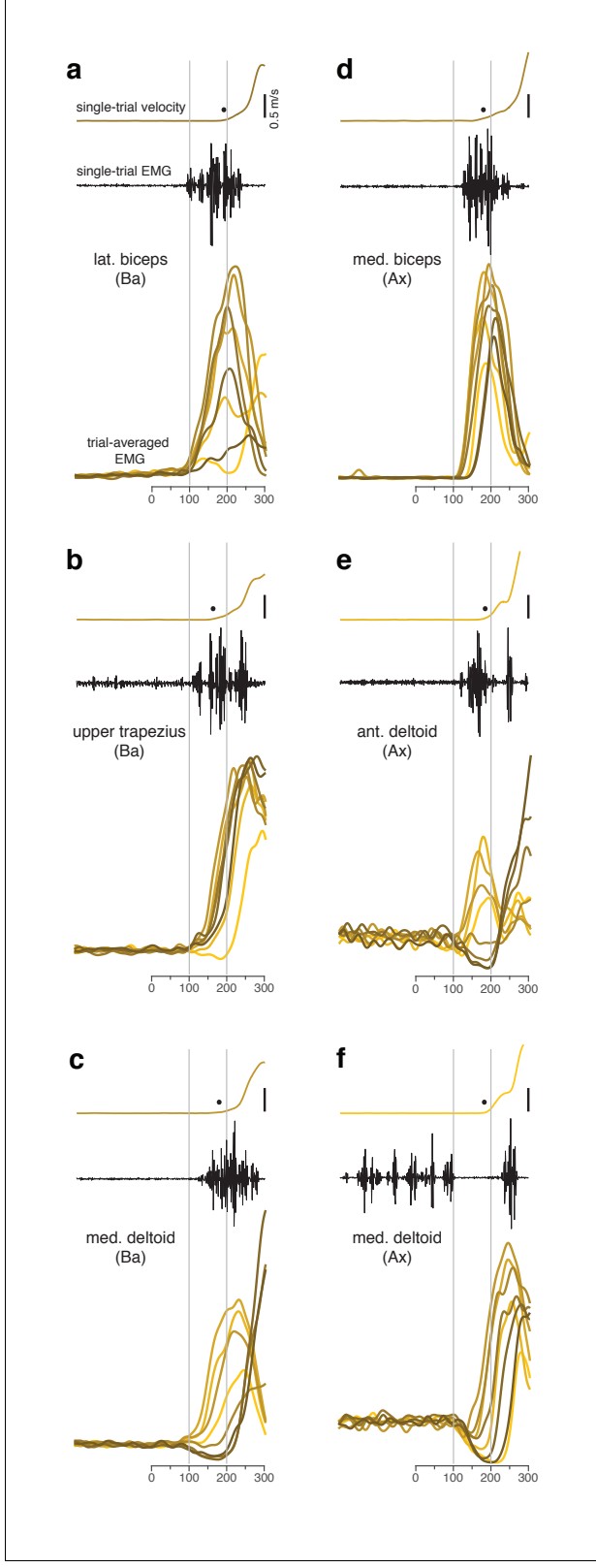

**Figure 3.** Muscle activity analyzed relative to target onset during the quasi-automatic context. (a–f) Each panel plots data for one muscle. Single-trial examples of hand velocity (*yellow*) and EMG (*black*) are shown at the top of each panel. *Filled circle* indicates the estimated time of movement onset on that trial. Average rectified-and-filtered EMG is shown, for each target direction, at the bottom of each panel. Averages were made locked to

*Figure 3 continued on next page*

*Figure 3 continued*

target onset. Thus, the first change provides an estimate of how soon EMG could change on trials with faster RTs. Filtering used a 10 ms Gaussian to minimize impact on latency. Shade-coding indicates reach direction as in *Figure 2*. Gray vertical lines mark 100 ms and 200 ms after target onset. Note that the target-locked averages used here do not use concatenation and are most appropriate for examining latency. Subsequent analyses employ concatenated averages, with movement-related activity locked to movement onset, which is more appropriate when examining activity patterns.

DOI: https://doi.org/10.7554/eLife.31826.007

The following figure supplement is available for figure 3:

**Figure supplement 1.** Responses from three muscles of the upper arm.

DOI: https://doi.org/10.7554/eLife.31826.008

and 3.0%) relative to the cue-initiated context, consistent with slightly lower kinematics for some reach directions (visible upon close inspection of *Figure 2e,f*).

Neural activity is most naturally interpreted as preparatory if there are not concurrent changes in muscle activity. This is pertinent for the cue-initiated and self-initiated contexts, where the target is known for an extended period before movement begins. As desired, there was little change in baseline EMG during that interval (*Figure 3—figure supplement 1*). For the cue-initiated context, the across-condition variance of EMG (i.e. selectivity for direction) during the delay was 0.76% and 2.8% of the variance during movement. For the self-initiated context, corresponding values were 0.51% and 1.2% (measured during the interval from target onset until 250 ms before movement onset).

## Single-neuron firing rates

Spikes of well-isolated neurons were recorded from the arm region of motor cortex, including M1 and the immediately adjacent region of PMd (129 and 172 neurons for monkey Ba and Ax). Spike-trains were filtered and trial-averaged to yield an estimate of firing rate versus time. Filtering employed a narrow Gaussian kernel (20 ms SD) to ensure preservation of response patterns that were often complex and multi-phasic (*Churchland and Shenoy, 2007b*). A shorter, 10 ms Gaussian kernel, was used for analyses of absolute latency (e.g. comparing neural and EMG onset latencies).

For visualization purposes, we desired a continuous estimate of firing-rate spanning target-driven and movement-related events. A continuous trace helps visually link events through time and is natural when plotting neural trajectories in state space. To achieve this, we concatenated target-aligned and movement-aligned epochs before filtering. The time of concatenation (*gray symbols* in *Figure 5*) was chosen based on behavioral performance, so that the separation between target onset and movement onset was close to typical. Doing so produces a representative trace and minimizes the small discontinuity in firing rate at the time of concatenation.

## Neural responses during the cue-initiated context

Many neurons showed activity that varied with reach direction during the delay-period of the cue-initiated context (e.g. *Figure 5*, *red traces*). For analysis, we defined a 450 ms 'delay epoch' beginning 50 ms after target onset. Delay-epoch activity varied significantly with reach direction (ANOVA, $p<0.05$) for the majority of neurons (74/129 and 88/172 for monkey Ba and Ax). We defined a 300 ms 'movement epoch', beginning 50 ms before movement onset and ending just after the hand landed on the target. Movement-epoch activity also varied significantly with target direction for most neurons (116/129 and 144/172). These epochs are used below when identifying putatively preparatory and movement-related subspaces. We chose epochs that were conservative; the delay epoch ended well before movement onset, and the movement epoch began just after EMG began changing, by which point any preparatory process has presumably completed.

We refer to delay-period activity as 'putatively preparatory' but stress that hypotheses differ regarding this interpretation. We also note that it is unclear exactly when putatively preparatory activity transitions to movement-related activity. Presumably this must happen following the go cue but before movement onset. Yet the moment of this hypothesized transition is difficult or impossible to determine via inspection. For example, the neuron illustrated in *Figure 5a* shows multiple response phases following the go cue, including a peak ~75 ms before movement onset (higher for rightwards reaches; *lighter red traces*) and a subsequent peak during movement (higher for

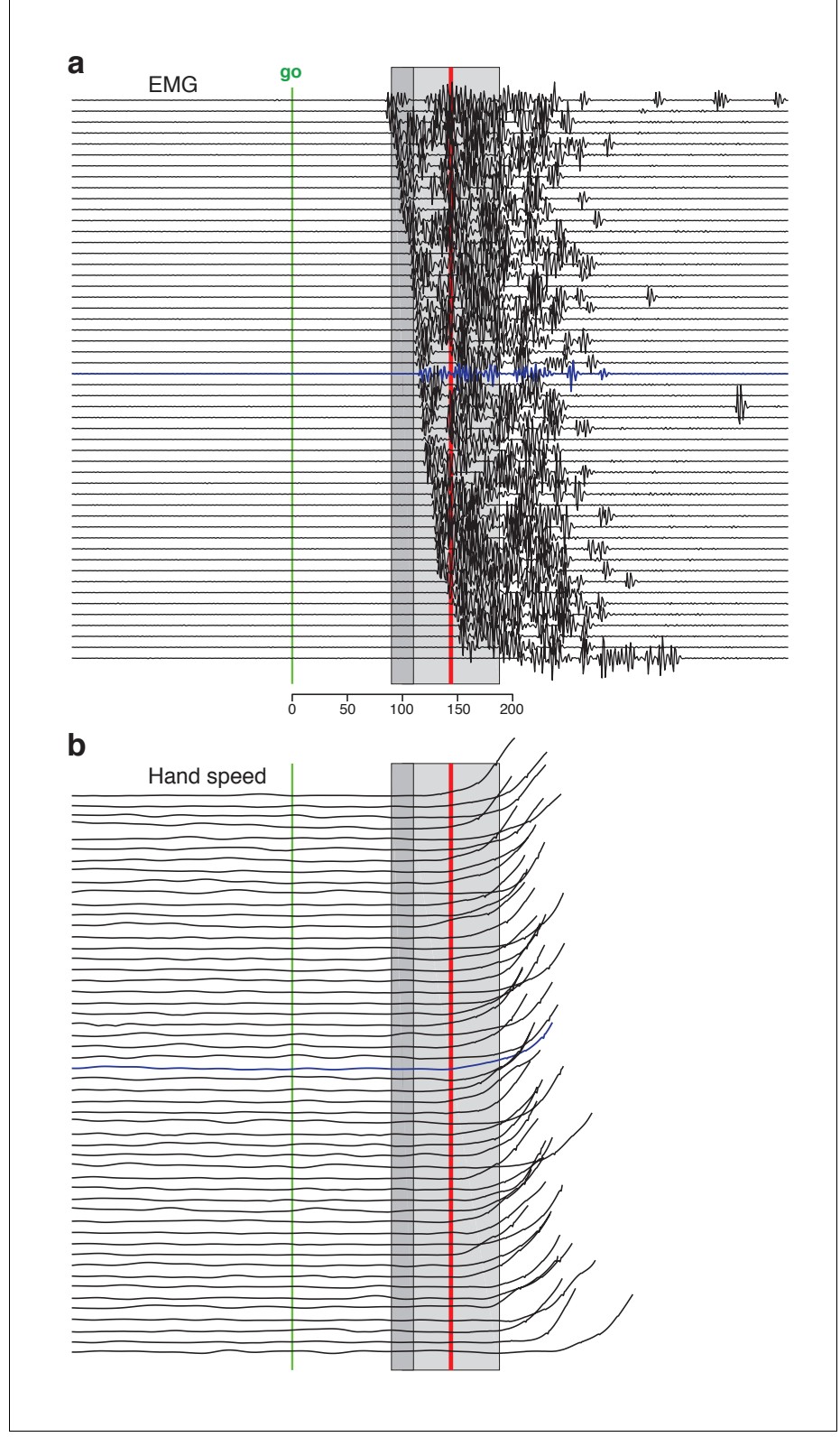

**Figure 4.** Single-trial EMG voltages during the quasi-automatic context, ordered by response latency. EMG data were recorded from the trapezius of monkey Ba. This recording (different from that in *Figure 3b*) had low baseline activity, making it possible to estimate EMG onset on individual trials. (a) Each trace plots the voltage recorded on a single trial. Ordering is based on estimated EMG onset latency. *Blue trace* indicates the trial with the median

*Figure 4 continued on next page*

*Figure 4 continued*

latency. EMG onset was estimated via inspection; trials were not analyzed if baseline EMG was not essentially zero. This was critical because fluctuations in a non-zero baseline produced ambiguity when assessing EMG onset. Analysis included trials for three reach directions that evoked the largest response. Shaded regions allow comparison with EMG latencies in *Perfiliev et al. (2010)*. For that study, the earliest EMG was observed at 90–110 ms (range is across subjects) and the mean onset of EMG was 144 (averaged across trials and subjects). These benchmarks are indicated by the *dark gray shaded region* and the *red line* respectively. The estimated typical within-subject range of latencies for Perfiliev is indicated by the *light gray shaded region*. This was estimated as symmetric about the across-subject mean, beginning at the average earliest latency (thus spanning from 44 ms below to 44 ms above the mean). (b) Hand-speed traces for the same trials as in panel a, using the same ordering. Traces are truncated to allow magnification of early events.

DOI: https://doi.org/10.7554/eLife.31826.009

The following figure supplements are available for figure 4:

**Figure supplement 1.** Same as *Figure 4* but for the biceps of monkey Ba.
DOI: https://doi.org/10.7554/eLife.31826.010
**Figure supplement 2.** Same as *Figure 4* but for the biceps of monkey Ax.
DOI: https://doi.org/10.7554/eLife.31826.011

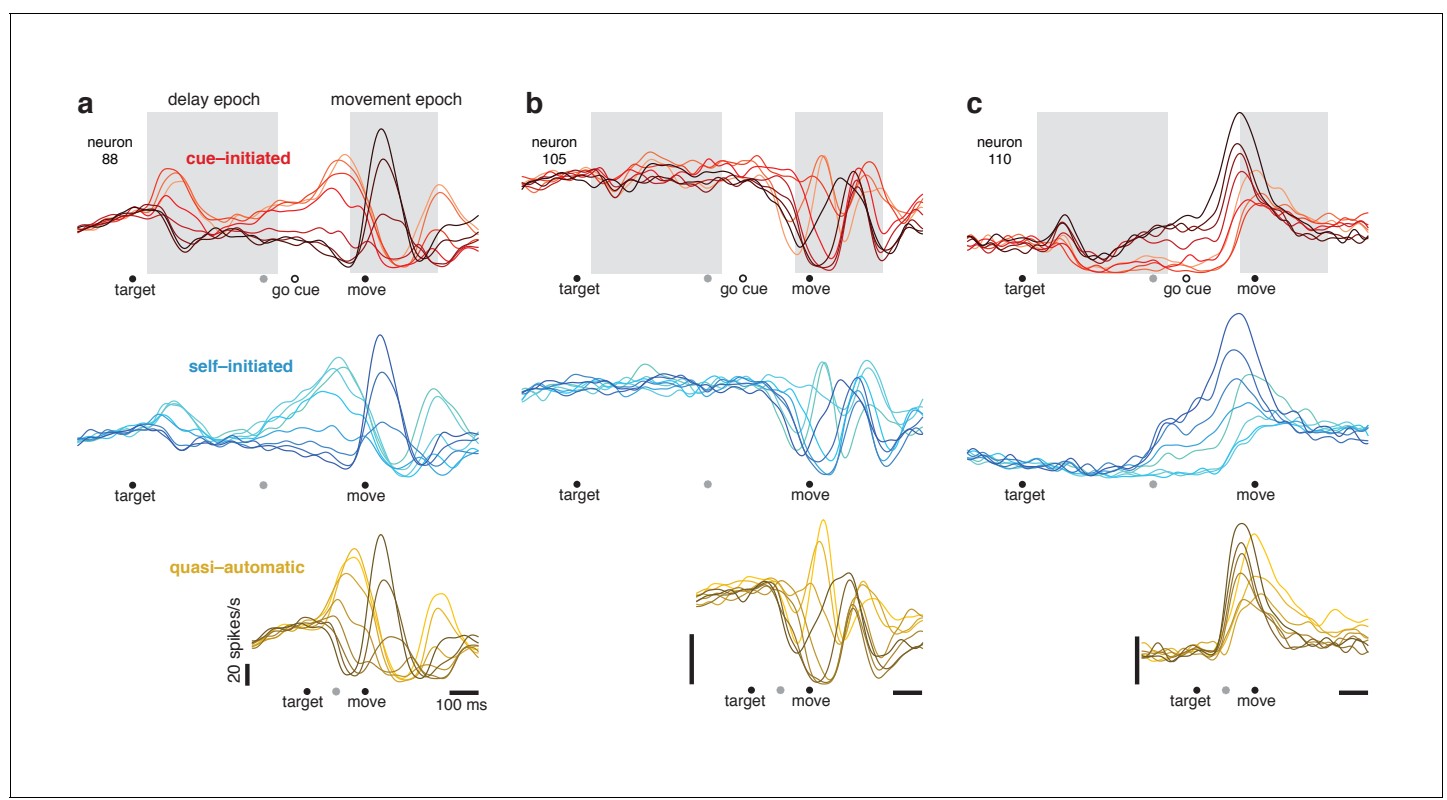

**Figure 5.** Responses of three example neurons. Each column shows the responses of a single neuron for the three contexts. Each trace plots the trial-averaged firing rate for one reach direction (same color scheme as in *Figure 2*). Gray shaded regions indicate delay and movement epochs, used to define the preparatory and movement dimensions in subsequent analyses. All traces are based on data that were aligned to target onset for the left-hand side of the trace, and to movement onset for the right-hand side of the trace. Concatenation occurred at the time indicated by the *gray symbol*. For the cue-initiated context, the left-hand side contains data from −200–450 ms relative to target onset (only trials with delays > 400 ms were analyzed). The right-hand side contains data from −350 ms before movement to 400 ms post movement. The indicated time of the go cue is based on the mean RT. The self-initiated context employed the same intervals as the cue-initiated context, to aid visual comparison. For the quasi-automatic context, the first 100 ms of the response is aligned to the target onset and the subsequent response is aligned to movement onset. All vertical calibration bars indicate 20 spikes/s. To aid comparison, all three examples are from the same monkey (Ba).
DOI: https://doi.org/10.7554/eLife.31826.012

leftwards reaches; *darker red traces*). Should the first peak be interpreted as a final strengthening of a preparation-related response or the beginning of a movement-related response? Is such a distinction even meaningful? Similar ambiguity was present even when neurons had seemingly simple responses. The neuron illustrated in *Figure 5c* (*red traces*) exhibits activity just before movement that is a magnified version of delay-period activity. Does that peak reflect the culmination of putatively preparatory activity, or is it a movement-related burst? These uncertainties highlight a known limitation of the instructed-delay paradigm: it is challenging to interpret neural events between the go cue and movement onset (*Crammond and Kalaska, 2000*). This challenge will become particularly relevant when examining responses during the quasi-automatic context, where that interval is the only opportunity for preparatory events.

## Single-neuron responses during the self-initiated context

If delay-period activity is primarily suppressive, then in the self-initiated context, delay-period-like activity should be present but should wane as movement approaches and suppression is relieved. The opposite should occur if delay-period activity is primarily preparatory: delay-period-like activity should grow with time in the self-initiated context. Single-neuron responses followed the second prediction. Pre-movement activity in the self-initiated context grew stronger with time and came to resemble delay-period activity in the cue-initiated context. For example, in *Figure 5c*, the ordering of traces ~ 250 ms before movement onset is similar for the self-initiated (*blue*) and cue-initiated (*red*) contexts.

During the first 150 ms after target onset, the median correlation between self-initiated and cue-initiated activity patterns was low (across-neuron median $r = 0.39$ and $0.16$ for the two monkeys), largely because activity was often weak in the self-initiated context. Correlations became stronger as movement onset neared (median $r = 0.86$ and $0.74$, using a 150 ms window ending 50 ms before movement onset). During movement, the correlation was quite high (median $r = 0.92$ and $0.87$).

## Single-neuron responses during the quasi-automatic context

Movement-related responses in the quasi-automatic context closely resembled those in the other two contexts (compare across rows in *Figure 5*). The median correlation between quasi-automatic and cue-initiated movement-epoch activity patterns was $0.85$ and $0.85$ (monkey Ba and Ax). The primary difference was a tendency for some features to be slightly magnified in the quasi-automatic context (e.g. the central peak in *Figure 5b*), consistent with the slightly greater reach speed and slightly stronger muscle activity in the quasi-automatic context.

Do responses in the quasi-automatic context contain putatively preparatory aspects? Addressing this question requires interpreting events between target onset and movement onset. Consider the neurons illustrated in *Figure 5a,c*. For both, the first pattern of activity to emerge in the quasi-automatic context resembled that shortly before movement onset in the cue-initiated context. But does activity at that time reflect the culmination of preparation or the beginning of movement-related activity? As discussed above, this is difficult to infer from individual-neuron responses.

## Segregating preparatory and movement responses at the population level

We employed a recently developed method to segregate the population response into putatively preparatory and movement-related aspects. This method leverages the unexpected observation that neuron-neuron correlations change dramatically between delay and movement epochs (*Elsayed et al., 2016*). Specifically, the 'neural dimensions' that best capture delay-epoch activity do not capture movement-epoch activity and vice versa. Our strategy is to use the cue-initiated context to identify a set of neural dimensions capturing delay-epoch activity and an orthogonal set of dimensions capturing movement-epoch activity. If delay-epoch activity reflects a standardly occurring preparatory process, the corresponding dimensions may similarly capture preparatory processing in the other two contexts. Alternatively, if delay-epoch activity reflects some non-preparatory process specific to an experimenter-imposed delay, the corresponding dimensions will either capture little activity during the other contexts or will capture activity unrelated to delay-epoch activity.

The simplest method for identifying putatively preparatory and movement-related dimensions is to apply principal component analysis separately to delay-epoch and movement-epoch data.

Empirically, this results in two sets of dimensions that are naturally close to orthogonal. We improved upon this approach by identifying all dimensions jointly (*Elsayed et al., 2016*); our cost function seeks one set of dimensions capturing maximal delay-epoch variance and an orthogonal set of dimensions capturing maximal movement-epoch variance. This achieves a result similar to principal component analysis but ensures all dimensions are truly orthogonal. Dimensions were found based on neural responses during the cue-initiated context only. We identified twelve dimensions, collectively the putative 'preparatory subspace', that captured 80% of delay-epoch firing rate variance but only 3% of movement-epoch variance. We isolated twelve further dimensions, collectively the 'movement subspace', that captured 85% of movement-epoch variance but only 3% of delay-epoch variance. The above percentages are for monkey Ba and were similar for monkey Ax (72% versus 4% and 83% versus 3%). We specifically sought dimensions capturing response aspects that vary with condition (i.e. are 'tuned' for reach direction). Untuned response aspects exist in separate, largely orthogonal, dimensions (*Kaufman et al., 2014*) and are not analyzed here.

We projected the population response onto each preparatory and movement dimension (*Figure 6*, *middle*). Each projection is a weighted sum of single-neuron responses (weights are the elements of the vector defining the dimension). Because weights are optimized to capture response

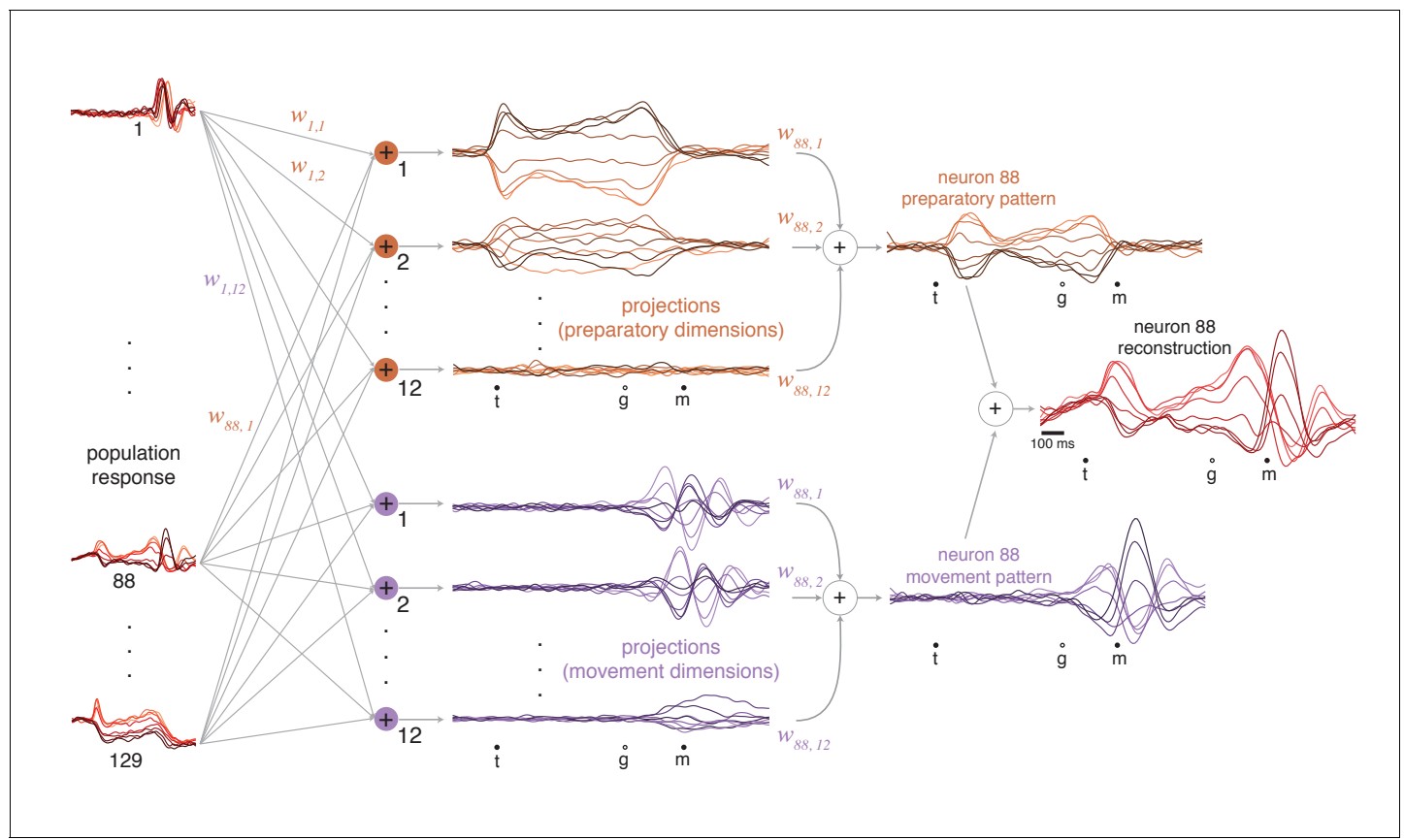

**Figure 6.** Illustration of how preparatory and movement-subspace projections were found using the cue-initiated context. The population of neural responses (*left column*) can be linearly 'read out' into preparatory and movement-related projections (*middle column*). $w_{n,k}$ is the readout weight from the $n^{th}$ neuron to the $k^{th}$ projection, and the collection of $w_{:,k}$ such weights is the $k^{th}$ neural dimension. Because the identified dimensions maximize the variance captured, individual-neuron responses can be reconstructed from the projections. Reconstruction employs the same weights that defined the projections. For example, if neuron 88 contributed to the first preparatory projection with weight $w_{88,1}$, then that first preparatory projection contributes to the reconstruction of neuron 88 with the same weight. The weighted sum of the preparatory projections yields that neuron's 'preparatory pattern' (*orange traces in right column*). The weighted sum of movement projections yields that neuron's 'movement pattern' (*purple traces in right column*). The full reconstruction is the sum of these two patterns, which describe the tuned aspects of the neuron's response, plus a time-varying mean that captures any untuned trends in the overall mean firing rate with time (not shown). The success of the reconstruction can be appreciated by comparison with the true response in *Figure 5a*.

DOI: https://doi.org/10.7554/eLife.31826.013

variance, projections are not only readouts but also building blocks of single-neuron responses (as for principal component analysis). Using these building blocks, it becomes possible to estimate different contributions to each neuron's response. For example, the response of neuron 88 (*Figure 6*, *rightmost column*) is accurately approximated as the sum of a preparatory-subspace pattern (a weighted sum of preparatory projections, *orange*) and a movement-subspace pattern (a weighted sum of movement projections, *purple*). The reconstruction is quite good ($R^2$ = 0.93, compare with *Figure 5a*, *top*), reflecting the high proportion of variance captured in the two subspaces.

## Reconstruction of neural responses across contexts

The subspaces found using the cue-initiated context continued to capture a high percentage of response variance in the self-initiated and quasi-automatic contexts. In the self-initiated context, the total variance captured across the two subspaces was 89% (monkey Ba) and 84% (monkey Ax) of that in the cue-initiated context. In the quasi-automatic context, the total variance captured was 87% and 84% of that in the cue-initiated context. The high percentage of variance captured is reflected in the accurate reconstruction of neural responses. For example, the response of neuron 88 (*Figure 5a*) was accurately reconstructed in both the self-initiated (*Figure 7a*) and quasi-automatic (*Figure 7b*) contexts.

Importantly, although the preparatory subspace was identified based only on the cue-initiated context, it also made contributions in the other two contexts. For example, the reconstruction for neuron 88 involved a robust preparatory-subspace pattern during the self-initiated context (*Figure 7a*, *orange*) and a short-lived but strong preparatory-subspace pattern during the quasi-automatic context (*Figure 7b*, *orange*). Within these patterns, the ordering of conditions was similar for all contexts: positive for rightwards reaches (*light orange traces*) and negative for leftwards reaches (*dark orange traces*). To ask if ordering is generally preserved across contexts, for each neuron we measured the preparatory-subspace pattern 100 ms before movement onset, yielding a vector with one value per direction. We regressed this vector for the self-initiated and quasi-automatic contexts against that for the cue-initiated context. If preparatory-subspace patterns are the same, regression will yield a slope of one. In contrast, a slope of zero would indicate no consistent relationship.

Slopes were strongly positive (*Figure 7c,d*, *black bars*) indicating that preparatory-subspace activity varies with reach direction in a similar way for all contexts. For monkey Ba, the slope was slightly greater than unity when comparing the quasi-automatic and cue-initiated contexts, consistent with quasi-automatic activity being slightly stronger. For monkey Ax, the slope was less than unity when comparing the self-initiated and cue-initiated contexts, consistent with self-initiated activity being either modestly weaker or slightly different. This will be explored further below. A subtle concern is that high similarity might arise trivially if neurons have similar directionality at all times. This was not the case. For example, there was no consistent relationship between the preparatory-subspace pattern in the cue-initiated context and the movement-subspace pattern in the other two contexts (*gray bars*).

## Temporal evolution of responses in preparatory and movement subspaces

At each time, we measured the across-condition variance (the strength of selectivity) of preparatory-subspace activity. This 'preparatory-subspace occupancy' reflects the size of the envelope describing the *orange* patterns in *Figures 6* and *7*. We similarly computed movement-subspace occupancy. Movement-subspace occupancy exhibited a similar time-course for all three contexts (*Figure 8*, *purple traces*): occupancy was negligible until ~100 ms before movement onset and peaked just after movement onset. In contrast, preparatory-subspace occupancy (*orange traces*) showed both differences and commonalities across contexts.

In the cue-initiated context, occupancy rose sharply following target onset and was sustained at a lower level throughout the delay (*Figure 8a*). Occupancy declined rapidly just before movement onset, reaching near-baseline levels around the time movement began (flanking traces show SEMs; see *Table 2* for key p-values). For monkey Ba, there was a prominent pre-movement peak. For monkey Ax, the pre-movement peak was more modest. Because data are normalized, this does not imply

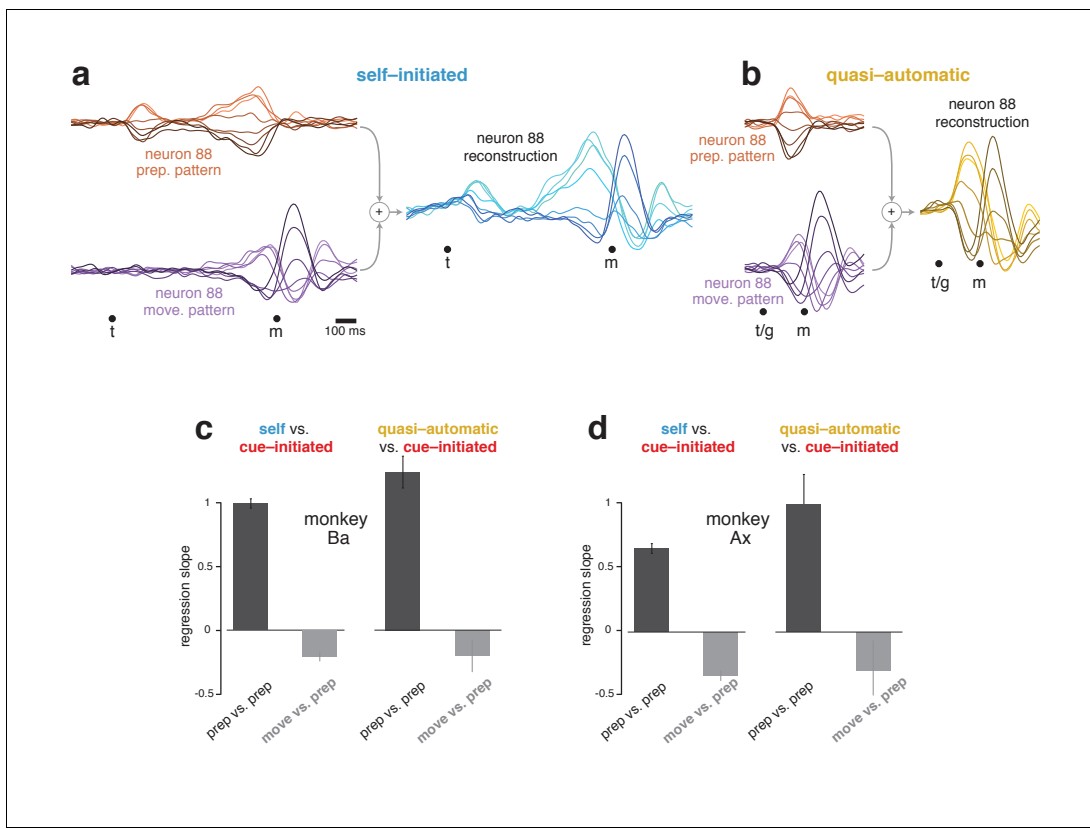

**Figure 7.** Reconstruction of single-neuron responses in the self-initiated and quasi-automatic contexts. (a) Example reconstruction for the self-initiated context. The population response was projected onto the preparatory and movement dimensions, found using the cue-initiated context. We then used those projections to reconstruct the preparatory (*orange*) and movement (*purple*) patterns that contributed to each neuron's response. These patterns are shown for neuron 88. The full reconstruction (*blue*) is the sum of the preparatory and movement patterns, plus the across-condition mean response (which has no tuning, but captures the overall mean rate at each time). This reconstructed response can be compared with the true response in *Figure 5a*. (b) As in (a), but for the quasi-automatic context. (c) Analysis of whether a neuron's preparatory-subspace pattern exhibits similar 'tuning' across contexts. For each neuron, we took the value of the preparatory subspace pattern, 100 ms before movement onset, for each direction. These values form a vector with one element per direction. To compute the similarity of the vector for the self-initiated context with that for the cue-initiated context, we regressed the former versus the latter and took the slope. The same was done for the quasi-automatic context versus the cue-initiated context. Dark bars show the average slope across neurons ± SEM. As a comparison, we repeated this analysis but regressed the movement pattern during the self-initiated and quasi-automatic contexts versus the preparatory pattern during the cue-initiated context (*light gray bars*). The movement pattern was assessed 150 ms after movement onset. Data are for monkey Ba. (d) As in (c), but for monkey Ax.
DOI: https://doi.org/10.7554/eLife.31826.014

that occupancy just before movement onset was small in absolute terms – merely that it was not as large as the post-target peak.

In the self-initiated context, preparatory-subspace occupancy developed more sluggishly (*Figure 8b*). The post-target peak was significantly weaker than for the cue-initiated context. Occupancy remained relatively weak for the next few hundred milliseconds. Preparatory-subspace occupancy then grew as movement approached, reaching a peak shortly before movement onset. By 150 ms before movement onset, preparatory-subspace occupancy was similar for the self-initiated and cue-initiated contexts: slightly greater for monkey Ba ($p<0.003$) and slightly smaller for monkey Ax (N.S.). The rise in preparatory subspace occupancy as movement approaches is consistent with the hypothesis that preparatory-subspace activity is preparatory. Were it suppressive, activity should wane as movement becomes more probable.

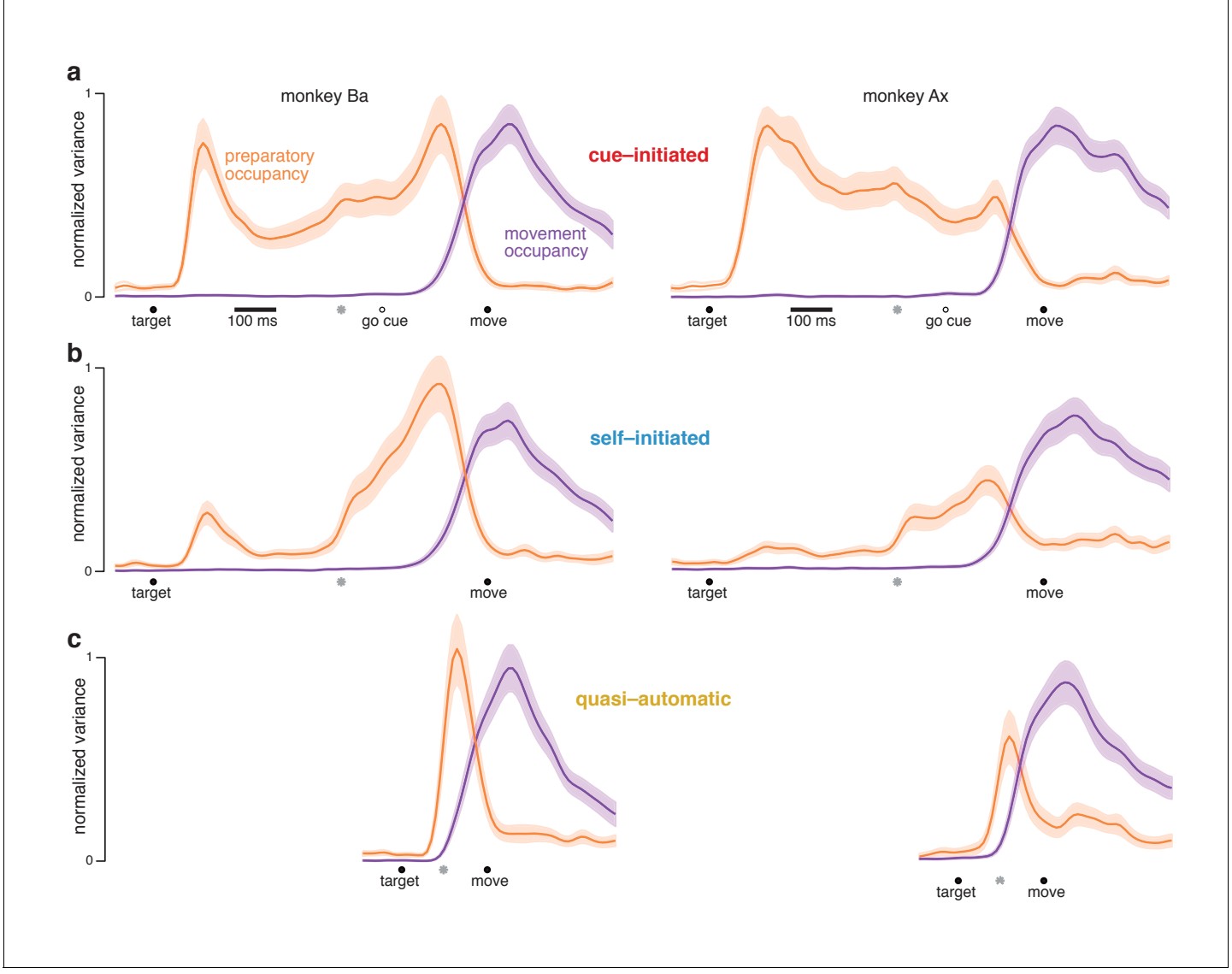

**Figure 8.** Preparatory and movement–subspace occupancy. (a) Preparatory and movement subspace occupancy for the cue–initiated context. The two columns show results for monkeys Ba and Ax. Preparatory-subspace occupancy, across all three contexts, was normalized by the highest value attained in the cue-initiated context. Movement-subspace occupancy was similarly normalized. The shaded region denotes the standard deviation of the sampling error (equivalent to the standard error of the mean) computed via bootstrap. *Gray symbols* indicate the time when target-locked data and movement-locked data were concatenated. Open circle indicates the average time of the go cue relative to movement onset. (b,c) Occupancy during the self-initiated and quasi-automatic contexts respectively.

DOI: https://doi.org/10.7554/eLife.31826.015

The following figure supplement is available for figure 8:

**Figure supplement 1.** Testing predictions of different hypotheses regarding the development of preparatory activity on single trials in the self-initiated context.

DOI: https://doi.org/10.7554/eLife.31826.016

Rising preparatory-subspace occupancy in the self-initiated context could reflect steady ramping on individual trials (with short RTs associated with a steeper ramp) or a more stereotyped onset at a variable time. Our simultaneous recordings were not sufficiently extensive to temporally resolve occupancy on individual trials. Nevertheless, some inferences can be made by separately analyzing long-RT and short-RT trials with data locked to movement onset (*Figure 8—figure supplement 1*). Results are inconsistent with trial-to-trial RT variability resulting from variable-slope ramping activity.

Results are more consistent with occupancy developing in a stereotyped manner (either a step or a rapid ramp) a few hundred milliseconds before movement onset.

In the quasi-automatic context, preparatory-subspace occupancy rose rapidly following target onset and peaked shortly before movement onset. Peak occupancy was slightly but significantly greater than the pre-movement peak in the other two contexts (p<0.002 for all comparisons). These findings are consistent with the hypothesis that a preparatory stage, involving delay-period-like activity, is present during the quasi-automatic context despite the absence of a delay and the low latency of the reaches. However, that putatively preparatory stage appears to be very rapid: preparatory-subspace occupancy precedes movement-subspace occupancy by only a few tens of milliseconds: 33 ms for monkey Ba and 42 ms for monkey Ax (assessed using a threshold of 10% of the peak, p<0.001 for both).

## The preparatory subspace is occupied for short-RT quasi-automatic trials

A small percentage of quasi-automatic trials had relatively long RTs and thus less time pressure (bottom of *panel a* in *Figure 4* and associated figure supplements). Might preparatory-subspace occupancy in the quasi-automatic context be primarily due to such trials? This seems unlikely a priori: if most trials lacked preparatory-subspace activity, trial-averaged activity would show very low preparatory-subspace occupancy. We further addressed this issue by re-analyzing the data after excluding trials with long RTs. For the cue-initiated and quasi-automatic contexts, we divided trials into RTs shorter versus longer than the median. For each set of trials, we computed individual-neuron firing rates and projected the population response onto the same dimensions analyzed in *Figure 8*. For the cue-initiated context, short-RT trials (*Figure 9a*, *red traces*) exhibited the typical rough plateau of preparatory-subspace occupancy, followed by a peak just before movement onset. For the quasi-automatic context (*yellow traces*) short-RT trials continued to show a pre-movement surge in preparatory-subspace occupancy. Indeed, the pre-movement peak was actually slightly higher for short-RT quasi-automatic trials versus short-RT cue-initiated trials. Thus, we saw no evidence that preparatory-subspace occupancy was absent for short-RT quasi-automatic trials; occupancy without a delay was slightly higher than with a delay. Additional analyses, presented below, confirm that preparatory events are not absent or fundamentally different for short-RT quasi-automatic trials.

We also observed that, across all comparisons, long-RT trials tended to have slightly higher preparatory-subspace peaks compared to short-RT trials. This was true for both the cue-initiated and quasi-automatic contexts (all traces peak slightly higher in *panel b* versus the corresponding traces in *panel a*). While this effect runs counter to the expectation that longer RTs might be associated with lower activity, prior studies have documented that this relationship is absent or inconsistent (*Michaels et al., 2015*; *Churchland et al., 2006*; *Afshar et al., 2011*). In the context of the present study, a higher peak in preparatory-subspace occupancy for long-RT trials is unsurprising. Occupancy starts to rise following the cue to move. Occupancy then falls just before movement onset. Later movement onsets thus allow slightly more time for activity to grow before being truncated. Subsequent state-space analyses show this effect more directly and reveal that it is quite modest (it is magnified when considering occupancy, which quantifies variance).

## Preparatory and movement events in state space

To visualize events within the preparatory and movement subspaces, we projected activity at different times onto planes within those subspaces. *Figure 10* plots 'snapshots' of those projections. *Figure 10—figure supplement 1* shows similar snapshots for monkey Ax. Each trace plots the evolution of the neural state for one reach direction over a 150 ms period, beginning at the indicated time. Preparatory-subspace activity was quite low-dimensional for this task: the first two dimensions captured much more variance than subsequent dimensions. For example for monkey Ba, the third preparatory dimension captured only 14% as much variance as the first. Thus, the preparatory-subspace projections in *Figure 10a* give a reasonably complete view. Movement subspace activity was considerably higher dimensional: there were many dimensions with structure that was clearly not noise. The projections in *Figure 10b* thus yield only a partial view of movement-subspace events.

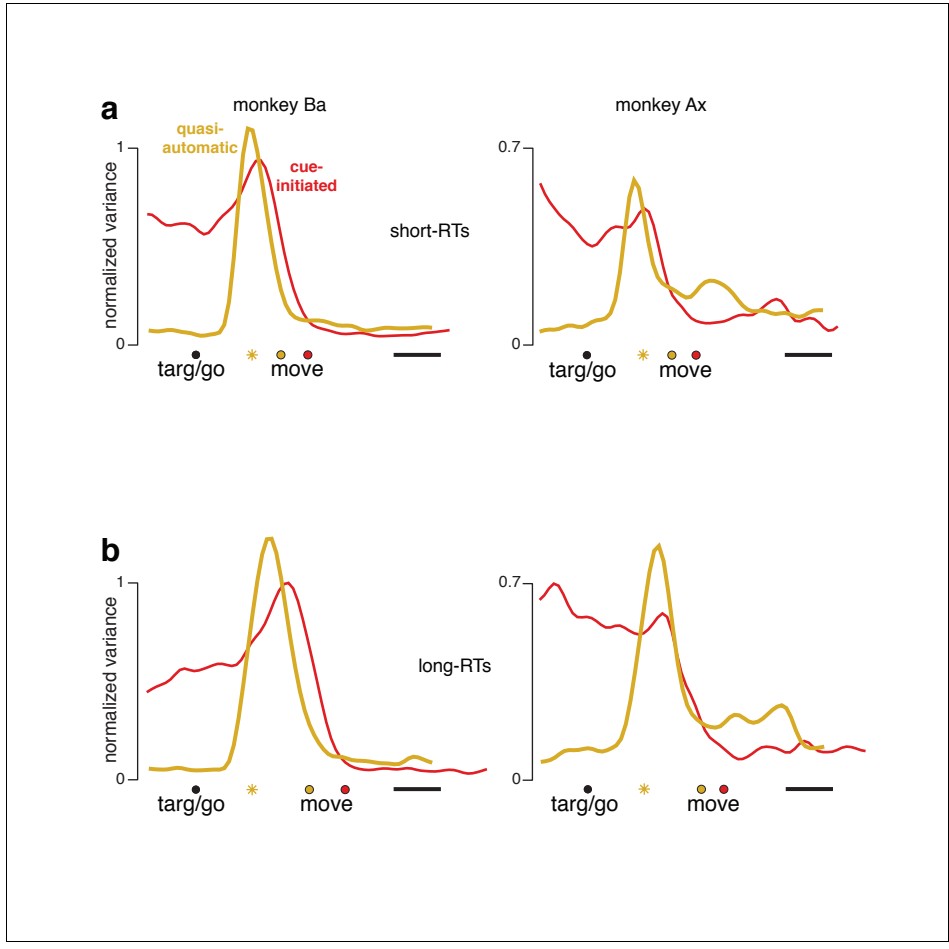

**Figure 9.** Preparatory-subspace occupancy for cue-initiated versus quasi-automatic contexts, after dividing trials by RT. (**a**) Data for trials with RTs shorter than the median. We computed the mean firing rate of each neuron for trials with RTs shorter than the median for that context. The population was then projected onto the same preparatory-subspace dimensions employed in *Figure 8*. Data are aligned so that target onset for the quasi-automatic context is aligned with the go cue for the cue-initiated context (black circle). Colored circles indicate the time of movement onset for the analyzed trials. This occurs earlier for the quasi-automatic context (yellow circle) relative to the cue-initiated context (red circle) due to the shorter RTs in the quasi-automatic context. Horizontal scale bar indicates 100 ms. Yellow star indicates the moment of concatenation for the quasi-automatic context. (**b**) Same analysis but for trials with RTs longer than the median. These longer RTs are reflected in the rightward-shifted time of movement onset relative to target onset/the go cue. The concatenated timespans have been adjusted to reflect the different separation between target and movement onset for short- and long-RT trials.
DOI: https://doi.org/10.7554/eLife.31826.018

Neural trajectories traced closely related paths for all contexts; the most notable differences regarded their timing. In the cue-initiated context, target onset prompted preparatory-subspace activity to become strongly selective for reach direction (*red traces* in *Figure 10a* separate upon target onset). The resulting pattern was sustained throughout the delay, then collapsed near the time of movement onset. In the self-initiated context, target onset prompted a weaker separation of preparatory-subspace neural states (*blue traces* in *Figure 10a* initially separate less than *red traces*). As movement onset approached, the preparatory-subspace pattern became more robust and closely resembled that in the cue-initiated context. For the quasi-automatic context, the preparatory-subspace pattern was very short-lived: it grew rapidly following target onset then immediately collapsed prior to movement onset. However, while present, the preparatory-subspace pattern during the quasi-automatic context closely resembled that in the other two contexts (compare across contexts in the third-to-last column of *Figure 10a*). For example, the dependence of the neural state on reach

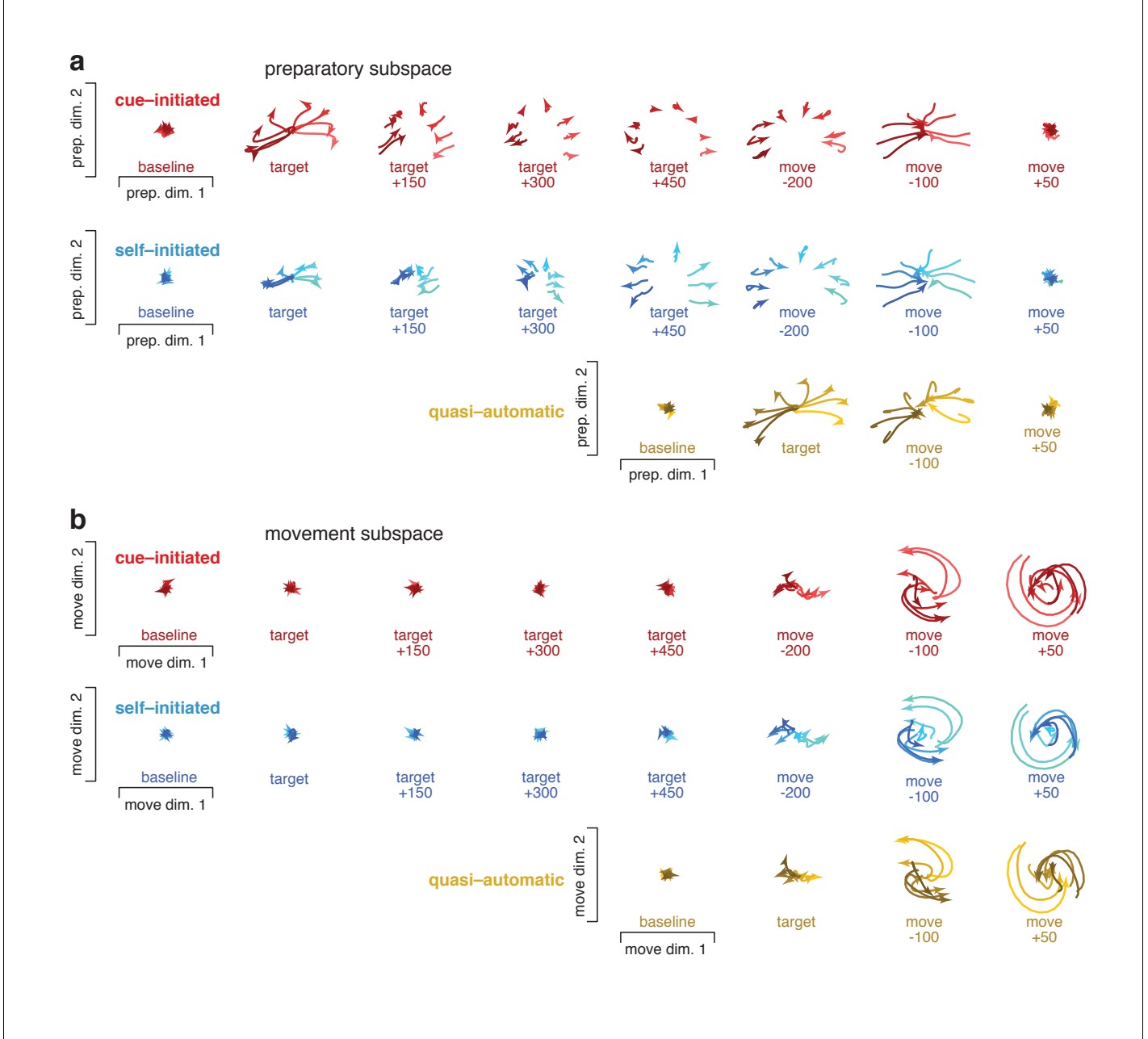

**Figure 10.** Snapshots of neural states in the preparatory and movement subspaces. Responses for cue–initiated (*red*), self–initiated (*blue*) and quasi–automatic (*yellow*) contexts projected onto the top two preparatory dimensions (a) and top two movement dimensions (b) Trace-shading corresponds to target direction (same color scheme as in *Figure 5*). Each snapshot shows the neural state in that subspace, for all eight directions, during a 150 ms window. Snapshots labeled 'baseline' begin 150 ms before target onset. Snapshots labeled 'target' plot data starting at target onset. For the cue–initiated and self–initiated contexts, the subsequent three snapshots show activity in 150 ms increments, still aligned to target onset. Snapshots labeled 'move −200' start 200 ms before movement onset, with data aligned to movement onset. Subsequent panels begin at the indicated time. Data are for monkey Ba.

DOI: https://doi.org/10.7554/eLife.31826.019

The following figure supplements are available for figure 10:

**Figure supplement 1.** Snapshots of neural states in the preparatory and movement subspaces for monkey Ax.
DOI: https://doi.org/10.7554/eLife.31826.020

**Figure supplement 2.** Snapshots of neural states comparing, for the quasi-automatic context, all trials with short-RT trials.
DOI: https://doi.org/10.7554/eLife.31826.021

direction was similar across contexts (*lighter*/*darker* traces indicate rightwards/leftwards movements).

Target onset produced essentially no separation of movement-subspace states for the cue-initiated or self-initiated contexts. Movement-subspace states started to separate ~100 ms before movement onset. The movement-subspace state then evolved according to rotational dynamics, as previously reported (*Churchland et al., 2012*) and in a manner predicted by neural network models (*Sussillo et al., 2015*) (see also *Hall et al., 2014*). As for such models, rotational dynamics were present in a subset of dimensions; the dimensions shown here were chosen to capture rotational dynamics for the cue-initiated context, and naturally capture similar dynamics for the other two contexts.

Comparing between subspaces reinforces and extends the results described in *Figure 8*. Across all contexts, preparatory-subspace activity always emerged before movement-subspace activity began. Preparatory-subspace activity and movement-subspace activity overlapped temporally: the former declined as the latter emerged. Just before and during that overlap, the pattern of preparatory-subspace activity was similar across contexts. As above, we also separately considered short-RT quasi-automatic trials. Events for short-RT trials were generally similar to those when considering all trials. However, as expected, the decline in preparatory subspace activity and the onset of rotational dynamics occurred slightly earlier for short-RT trials (*Figure 10—figure supplement 2*).

## Quantification of similarity across contexts

To quantify the similarity of movement-subspace events, we measured the correlation between the pattern in one context and that in another context. This analysis considered activity during a 150 ms window starting at movement onset. Correlations were high for all comparisons. Comparing self-initiated and cue-initiated contexts, correlations for the two dimensions shown in *Figure 10* were $0.96 \pm 0.01$ (95% CI) and $0.98 \pm 0.01$ for monkey Ba, and $0.98 \pm 0.01$ and $0.97 \pm 0.01$ for monkey Ax. Across the twelve movement dimensions, correlations ranged from 0.94 to 0.99 for monkey Ba and from 0.91 to 0.99 for monkey Ax ($p < 0.0001$ in all cases). Correlations were similarly high when comparing quasi-automatic and cue-initiated contexts. For the two dimensions shown in *Figure 10*, correlations were $0.96 \pm 0.02$ and $0.98 \pm 0.01$ for monkey Ba, and $0.95 \pm 0.02$ and $0.96 \pm 0.02$ for monkey Ax. Across all movement dimensions, correlations ranged from 0.94 to 0.98 for monkey Ba and from 0.88 to 0.98 for monkey Ax ($p < 0.0001$ in all cases).

A key question is whether these similar movement-subspace patterns are preceded by similar preparatory-subspace patterns. To address this question we focused on the preparatory subspace at a time – 55 ms before movement onset – when both preparatory and movement subspaces are occupied. At this time, preparatory-subspace occupancy is still high and movement-subspace activity has just started to emerge. It has been hypothesized that movement-generating dynamics are seeded by preparation (*Churchland et al., 2006*; *Elsayed et al., 2016*; *Churchland et al., 2010*; *Churchland et al., 2012*; *Churchland and Cunningham, 2014*; *Sussillo et al., 2015*; *Michaels et al., 2016*; *Hennequin et al., 2014*). Under this hypothesis, movement-related activity is largely determined by the preparatory state when movement-related activity begins. The preparatory-subspace state at that moment is thus predicted to be similar across contexts, given that subsequent patterns of movement-subspace activity are similar. We refer to that potentially critical preparatory-subspace state as the 'final' preparatory state. After that moment, preparatory-subspace activity declines to near-baseline levels and movement-subspace activity becomes strong.

The final preparatory state was similar across contexts. For each reach direction, the triplet of neural states for the three contexts formed a cluster (grouped via covariance ellipses in *Figure 11a, b*). Clusters were quite tight for monkey Ba and somewhat less so for monkey Ax. To quantify tightness of clustering – i.e. the similarity of states across contexts – we computed the correlation, between contexts, of the set of preparatory-subspace states (one state per reach direction). Considering the top two dimensions, the correlation between cue-initiated and self-initiated contexts was 0.99 for monkey Ba and 0.95 for monkey Ax (95% CIs were 0.98–0.99 and 0.86–0.98). The correlation between cue-initiated and quasi-automatic contexts was 0.99 for monkey Ba and 0.91 for monkey Ax (0.97–1.0 and 0.76–0.97). The correlation between self-initiated and quasi-automatic contexts was 0.99 for monkey Ba and 0.92 for monkey Ax (0.97–1.0 and 0.79–0.97). All correlations were statistically significant ($p < 0.001$). Results were similar if we considered all twelve preparatory dimensions. Although higher dimensions primarily captured noise, their inclusion did not have a large impact on correlations because they contributed little variance.: correlations ranged from 0.73 to

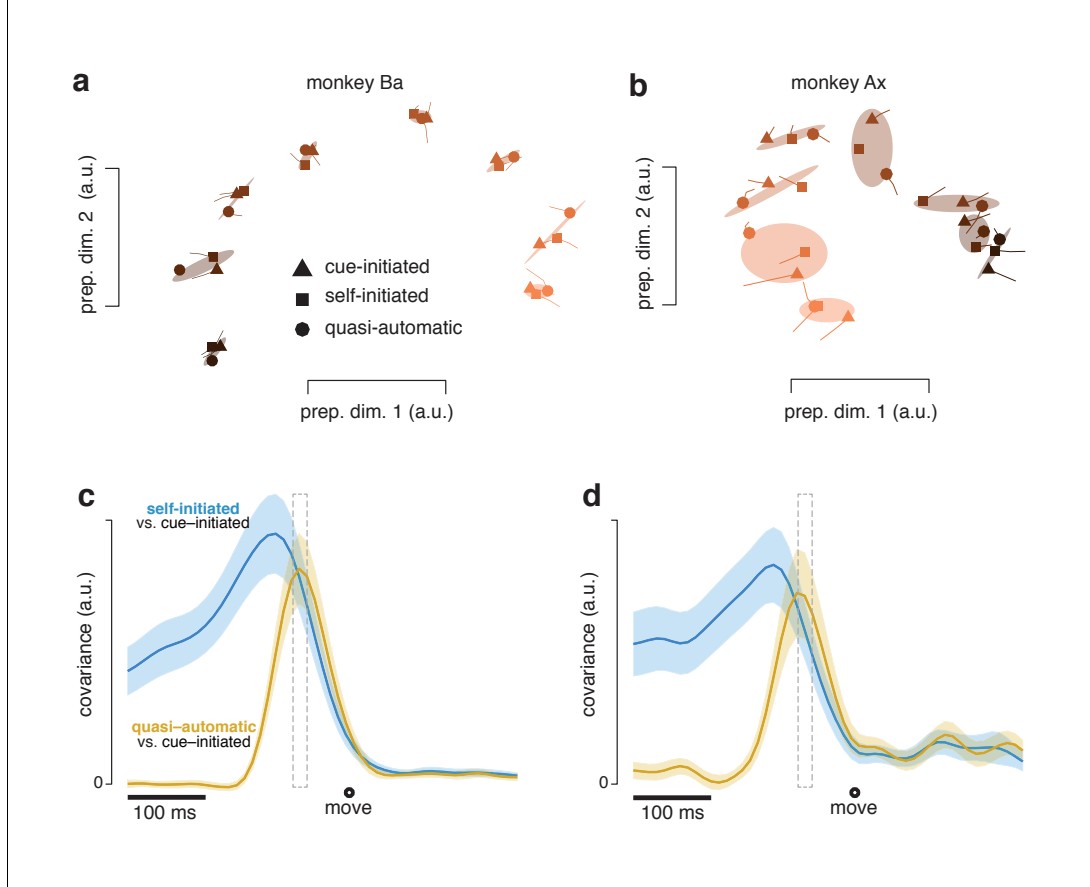

**Figure 11.** Preparatory subspace activity just before movement onset. (**a**) Data for monkey Ba. Each marker denotes the neural state in the two preparatory dimensions that captured the most variance, as in *Figure 10a*. Markers indicate the state 55 ms before movement onset. Tails plot 20 ms of activity leading up to that time. The three shapes show states for the three contexts. Shaded regions plot the covariance ellipse for each triplet of states. A different symbol shade is used for each target direction (*light* for right, *dark* for left). (**b**) As in (**a**) but for monkey Ax. (**c**) Quantification of the time-course of the similarity in the pattern of preparatory states between contexts. *Blue trace* plots the covariance between the preparatory pattern in the self-initiated context and that in the cue-initiated context. *Yellow trace* plots the covariance between the preparatory pattern in the quasi-automatic context and that in the cue-initiated context. The covariance is high when patterns are both strong and similar. The vertical scale is arbitrary (for reference, the correlation peaks close to one). Correlations are computed using all 12 preparatory dimensions. Results are very similar when considering only the top two dimensions. Gray dashed window of time indicates the 20 ms time range (from 75 to 55 ms before movement onset) shown in (**a**). The shaded regions denote the standard deviation of the sampling error (equivalent to the standard error) computed via bootstrap. Data are from monkey Ba. (**d**) As in panel c, but for monkey Ax.

DOI: https://doi.org/10.7554/eLife.31826.022

The following figure supplement is available for figure 11:

**Figure supplement 1.** Same analysis as in *Figure 11a,b* but after restricting analysis to trials with RTs shorter than the median.

DOI: https://doi.org/10.7554/eLife.31826.023

0.95 (across three comparisons for two monkeys), only modestly reduced from the 0.91–0.99 range for two dimensions. Correlations were intermediate (0.77–0.98) when measured in six dimensions.

The clustering of preparatory states, evident in *Figure 11a,b*, was very similar when considering trials with RTs shorter than the median (*Figure 11—figure supplement 1*). This confirms the results of *Figure 9*: preparatory-subspace activity remains robust even when excluding trials with longer RTs. For the analyses in both *Figure 11a,b* and *Figure 11—figure supplement 1*, clustering was very tight for monkey Ba (preparatory-subspace states were nearly identical across contexts) and less so for monkey Ax (preparatory-subspace states were quite similar, but not identical, across contexts). The less-tight clustering for monkey Ax may reflect the small across-context differences in the reaches being prepared and executed or may reflect a small influence of context itself on the preparatory state.

**Table 2.** Significance tests for features of the time-course of preparatory subspace occupancy.
P-values were computed via bootstrap. Neurons were redrawn and the central analysis (computing subspaces etc.) was performed again for the redrawn population. This process was repeated 1000 times, and the relevant measure was made, yielding a sampling distribution of 1000 values. P-values were based on these sampling distributions.

| | Monkey ba | Monkey ax |
|---|---|---|
| Prep. occ. declines before movement onset. (150 ms before vs. 150 ms after move onset, cue-initiated) | p<0.001 | p<0.001 |
| Target-driven rise in prep. occ. is smaller for self- vs. cue-initiated (compared 150 ms after target onset) | p<0.001 | p<0.001 |
| Self-initiated prep. occ. remains above baseline after post-target peak (400 ms post-target vs 150 ms pre-target) | p<0.001 | p<0.001 |
| Self-initiated prep. occ. grows as movement approaches (150 ms before move onset vs 150 ms after target onset) | p<0.001 | p<0.001 |
| Prep. occ. before movement onset is similar but slightly different for self- and cue-initiated contexts (assessed 150 ms before move onset) | p<0.003 (larger for self-) | N.S. (smaller for self-) |
| Prep. occ. rises above baseline in the quasi-automatic context | p<0.001 | p<0.001 |
| Pre-movement peak in prep. occ. is modestly higher for quasi-automatic context versus other two contexts | p<0.002 for both comparisons | p<0.002 for both comparisons |
| Prep. occ. precedes move. occ . even for quasi-automatic context (compare latency to cross threshold set to 10% of peak) | p<0.001 | p<0.001 |

DOI: https://doi.org/10.7554/eLife.31826.017

The above results establish that preparatory-subspace activity, shortly before movement onset, is similar across contexts. To assess the time-course of similarity, at each time we computed the covariance between the neural states in one context and those in another. Covariance reflects both similarity and strength and is thus expected to peak at a time when preparatory patterns are both similar and robust. When comparing the self-initiated and cue-initiated contexts (*Figure 11c,d*; *blue*) covariance rose as movement approached, peaking slightly more than 100 ms (monkey Ba and Ax) before movement onset. This is consistent with what can be observed in earlier figures: in the preparatory subspace, the pattern of states in the self-initiated context generally resembles the pattern in the cue-initiated context but is weaker until movement onset nears.

When comparing the quasi-automatic and cue-initiated contexts (*Figure 11c,d*; *yellow*) covariance rose rapidly, peaking ~65 ms and ~75 ms (monkey Ba and Ax) before movement onset. These peaks occur just after activity in the movement-subspace first begins to change, at ~75 ms (monkey Ba) and ~80 ms (monkey Ax) prior to movement onset in the quasi-automatic context. The narrowness of the peak underscores that there was only a brief period where both contexts had strong preparatory activity with similar structure. Thus, while the pattern of preparatory subspace activity in the quasi-automatic context comes to closely match that in the cue-initiated context, this similarity occurs late, just as movement-subspace activity is developing. This is consistent with a preparatory process that is present across contexts but unfolds very rapidly in the quasi-automatic context.

## Methodology and the segregation between preparatory and movement subspaces

Our dimensionality reduction approach maximizes the degree to which the preparatory subspace captures delay-period activity in the cue-initiated context, and the degree to which the movement subspace captures movement-related activity in the cue-initiated context. Importantly, this approach can segregate activity into separate subspaces only if the data naturally have that property. This is

not in general the case (*Elsayed et al., 2016*). For example, while recent dynamical models (*Churchland et al., 2012*; *Sussillo et al., 2015*) employ a null space in which preparatory activity can develop (*Kaufman et al., 2014*), they do not exhibit preparatory and movement-related activity that live in orthogonal dimensions. Segregation would also not have been possible if tuning were similar between preparation and movement. Thus, the almost complete disappearance of preparatory-subspace occupancy during the movement, and the almost complete absence of movement-subspace occupancy during the delay, are features of the data that were not appreciated until recently.

We used broad epochs of time to define the two subspaces and thus imposed no particular structure upon the time-course of occupancy. Because subspaces were identified blind to all data from the self-initiated and quasi-automatic contexts, the magnitude and time-course of occupancy in those contexts is a consequence not of the method but of shared aspects of the population response across contexts. The exact choice of temporal epochs used to define the subspaces generally had little impact on our results. However, we did find that such choices could subtly impact the profile of delay-period occupancy. For example, the initial peak in *Figure 8a* was somewhat more prominent if we shifted the delay-epoch earlier, and the final peak was somewhat more prominent if we shifted the delay-epoch later. This indicates that the subspace being occupied changes subtly over the course of the delay.

Dimensions found using cue-initiated data allow one to address whether events during the cue-initiated context also occur during the other contexts. Those dimensions do not address a different question: might there exist additional structure in the other contexts, outside the subspaces identified using the cue-initiated context? This is essentially guaranteed to be true to some degree but cannot be a large effect: preparatory and movement subspaces captured nearly as much variance (86% on average) for the self-initiated and quasi-automatic contexts as they did for the cue-initiated context. We also addressed this question by repeating the analysis in *Figure 8* but with preparatory and movement subspaces based on activity during the self-initiated context. Results were very similar overall, but preparatory-subspace occupancy in the self-initiated context was accentuated slightly for monkey Ba and noticeably for monkey Ax. This finding should be interpreted with some caution – dimensions will of course tend to best capture the data upon which optimization is based. Nevertheless, the result indicates that the subspaces occupied in the cue-initiated and self-initiated contexts are not perfectly aligned. This could indicate aspects of activity unique to self-initiated movements or may reflect the small differences in the upcoming reaches themselves. These same possibilities relate to the finding that there was a good but not always perfect match between the final preparatory states for the three contexts (*Figure 11a,b*).

## Relative timing of movement-related events

Given that motor cortex connects directly and indirectly to motoneurons, we wished to characterize the timing between movement-subspace occupancy and the onset of muscle activity. For both, we assessed latency by measuring the moment when activity surpassed 10% of its peak. To minimize the impact of filtering on latency, these analyses employed a 10 ms Gaussian filter for both neural and EMG data. Across monkeys and contexts, the movement subspace always became occupied just before the onset of changes in EMG, with an average latency of 21 ms. For comparison, the conduction delay from cortex to muscles, assessed via spike-triggered averages, can be as little as 6 ms from the time of a spike to the peak of the EMG response (*Fetz and Cheney, 1980*). This delay is slightly shorter (~4 ms for the lowest-latency neurons) when considering the initial rather than peak EMG response. Thus, activity in the movement subspace rises early enough to potentially account for the onset of muscle activity. This was consistently true across contexts, although with slight variability. The latency between the onset of movement-subspace activity and muscle activity was, for monkey Ba and Ax, 27 and 20 ms (cue-initiated), 33 and 22 ms (self-initiated) and 19 and 6 ms (quasi-automatic). These exact latencies should be interpreted with some caution: latencies are notoriously difficult to assess because high thresholds overestimate latency while low thresholds are sensitive to noise. Still, our best estimates indicate that, if cortico-motoneurons draw from movement-subspace activity, the onset of such activity occurs early enough to contribute to the onset of muscle activity.

## Discussion

### Is delay-period activity a reflection of motor preparation?

The first studies of delay-period activity employed two target directions and noted that directional selectivity could either agree with subsequent movement activity – suggesting a preparatory role – or disagree – suggesting a suppressive role (*Wise et al., 1986*). Later experiments leveraged a greater range of movements, revealing that delay-period and movement-related selectivity typically differ (*Crammond and Kalaska, 2000*; *Elsayed et al., 2016*; *Churchland et al., 2010*; *Kaufman et al., 2010*). Thus, delay-period activity is not a subthreshold form of movement-related activity. It was instead suggested that delay-period activity may initialize neural dynamics that will produce movement (*Churchland et al., 2006*; *Churchland et al., 2010*; *Churchland et al., 2012*; *Churchland and Cunningham, 2014*). Yet a recent study yielded mixed evidence for this hypothesis when comparing activity with and without a delay (*Ames et al., 2014*), highlighting the longstanding uncertainty regarding whether delay-period activity represents an essential preparatory process, a facilitatory but non-obligatory process, or a suppressive process specific to an artificial imposed delay (*Wong et al., 2015*).

The present results reveal that the neural process present during a delay-period is not specific to that situation. Delay-period-like activity is consistently observed in other contexts and displays the following properties. First, activity occupies a 'preparatory' subspace orthogonal to that occupied during movement. Second, activity in that preparatory subspace consistently occurs before movement-subspace activity. Third, activity in the preparatory subspace achieves a direction-specific state before movement onset, and this state is similar regardless of the presence of an imposed delay. Similarity is most robust at the critical moment when movement-subspace activity is just beginning.

These results do not prove that preparatory-subspace activity is preparatory, only that it follows the major predictions of that hypothesis. It is currently not feasible to perform a key causal test: perturbing activity in the preparatory subspace only. Still, non-specific disruption of premotor cortex activity impacts RT in a manner consistent with disruption of a preparatory process (*Churchland and Shenoy, 2007a*). Given prior findings and present observations, we tentatively interpret preparatory-subspace activity as preparatory and ask what conclusions might follow.

### Does preparation necessarily consume time?

Early behavioral studies both leveraged and supported the assumption that it takes considerable time to 'plan' or 'specify' the desired movement (*Rosenbaum, 1980*; *Ghez et al., 1997*; *Rosenbaum, 2010*). Subsequent physiology and modeling proposed that preparation involves the time-evolving shaping of neural activity specifying movement parameters (*Bastian et al., 2003*; *Bastian et al., 1998*; *Erlhagen and Schöner, 2002*) or the time-consuming convergence to a neural state that initializes movement-generating dynamics (*Michaels et al., 2015*; *Churchland et al., 2006*; *Elsayed et al., 2016*; *Churchland et al., 2010*; *Churchland et al., 2012*; *Churchland and Cunningham, 2014*; *Sussillo et al., 2015*; *Hennequin et al., 2014*). The view that preparation is time-consuming has enjoyed explanatory power and has motivated successful comparisons of trial-to-trial RT and neural variability (*Riehle and Requin, 1993*; *Michaels et al., 2015*; *Bastian et al., 2003*; *Churchland et al., 2006*; *Afshar et al., 2011*).

Yet there exist compelling recent arguments against the necessity of a time-consuming preparatory process (*Wong et al., 2015*; *Perfiliev et al., 2010*). The present findings support those arguments. In the quasi-automatic context, EMG latencies could be as short as 90–110 ms on some trials. Thus, there cannot exist an obligatory preparatory process that necessarily takes 100–200 ms to complete. Neural data are consistent with a preparatory process that can be very rapid. In the quasi-automatic context, preparatory subspace activity leads movement-subspace activity by only ~40 ms, and the preparatory-subspace state came to match that in the cue-initiated context in ~70 ms. These findings rule out the idea of a slow, cognitive planning process that must complete before movement begins. These findings support the hypothesis that preparatory activity is necessary to seed movement-generating dynamics but indicate that such activity can develop quite rapidly.

However, when time is available (as for both the self-initiated and cue-initiated contexts) the preparatory subspace becomes occupied well before movement onset. If preparation can be rapid, there would seem little advantage to this strategy – why not simply wait to prepare until just before

movement onset? We can only speculate, but the ability to rapidly and consistently achieve the correct preparatory state may be uncertain in many real-world situations. The motor system may thus have developed the strategy of preparing in advance when possible, allowing time to correct small inaccuracies before movement begins (*Churchland et al., 2006*; *Churchland and Shenoy, 2007a*). More broadly, rapid preparation may be feasible only for well-practiced movements, leading to a default strategy of preparing in advance when possible.

## Putatively preparatory and movement-related processes overlap

Preparatory-subspace activity declines as movement-subspace activity develops, with an overlap of ~100 ms. This overlap is consistent with (and indeed required by) the hypothesis that preparatory-subspace activity seeds movement-subspace dynamics. Aspects of the overlap explain a seeming discrepancy. (*Ames et al., 2014*) found that the neural state for zero-delay trials does not pass through the state achieved during the delay of long-delay trials. This seems at odds with our finding that a consistent preparatory-subspace state is achieved across all contexts. In fact, our results are fully compatible. Preparatory-subspace activity in the quasi-automatic and cue-initiated contexts became maximally similar slightly after movement-subspace activity emerged. Thus, at the moment when preparatory-subspace states are similar, the full neural state contains both preparation-related and movement-related contributions (and will also contain condition-independent components linked to movement initiation (*Kaufman et al., 2016*)). The full neural state at this moment will therefore not match that during a delay period, when only the preparatory-subspace contribution is present.

With this conflict resolved, the present results support and extend two key conclusions of Ames et al. First, the initial response to target onset is similar for zero-delay and long-delay trials (*Figure 10a*, compare the 'target' panel between cue-initiated and quasi-automatic contexts). Second, when under time pressure, the neural state does not momentarily pause at a stable state prior to the onset of movement-related activity (also see (*Michaels et al., 2015*)). Indeed, in the quasi-automatic context, events are so compressed that preparatory-subspace activity is still developing as movement-subspace activity is beginning.

## Stability of preparatory activity across the delay period

We have hypothesized that motor preparation involves achieving a neural state that will appropriately seed movement-period dynamics. In its most idealized form, this hypothesis predicts that an appropriate preparatory state should be achieved soon after target onset and should remain static across any imposed delay. Models incorporating this idealized assumption yield reasonably realistic single-unit responses (*Churchland et al., 2012*). More sophisticated network models (*Sussillo et al., 2015*; *Michaels et al., 2016*) produce similar but less-idealized responses: preparatory activity is relatively stable during the delay but is not perfectly static. Network models do not require perfect stability because multiple similar, but non-identical, preparatory states can produce nearly identical network outputs (as proposed by the 'optimal subspace hypothesis' (*Churchland et al., 2006*)).

The empirical population response during the cue-initiated context behaved much like that of network models. In the dominant preparatory dimensions, activity developed quickly and remained relatively stable across the delay. However, there were still some fluctuations with time, as can be seen in both state-space plots (*Figure 10*) and in projections as a function of time (*Figure 6*). Such fluctuations were also present in less-dominant dimensions and in 'untuned' response components (which were not analyzed here). Thus, the ideal of stable preparatory activity is only a first-order approximation, and this is true for both network models and the empirical data.

Very unstable activity during a memory period was recently observed in parietal cortex and ventral premotor cortex (*Michaels et al., 2018*). It is possible that neural events unfold differently during a memory period versus a delay period. However, a more interesting possibility is that the stability of preparatory activity may be area-specific. Activity may be more stable in areas where it serves a direct pre-motor purpose (initializing movement-period dynamics) and less so in areas where it relates to anticipation or more cognitive aspects of the task.

## Preparing versus deciding

Our data argue against the conception of preparation as an intrinsically slow, cognitive process. Yet our findings remain consistent with the hypothesis that slow cognitive processes influence preparatory activity. It is well established that pre-movement activity in a variety of brain regions can reflect decisions regarding when or where to move (*Thura et al., 2012*; *Cisek and Kalaska, 2005*; *Roitman and Shadlen, 2002*; *Kaufman et al., 2015*; *Guo et al., 2014*; *Murakami et al., 2014*). Such decisions can sometimes unfold slowly or vacillate with time. In motor cortex, preparatory subspace activity may therefore sometimes evolve slowly simply because the overall movement goal is being decided slowly.

The rising strength of preparatory-subspace activity in the self-initiated context is somewhat reminiscent of the rise of choice-related activity in decision tasks. However, in the present study, the target was always fully specified; target choice did not become more certain with time. Thus, rising preparatory-subspace activity presumably reflects preparation to execute a choice that was clear from the outset (also see (*Murakami et al., 2014*; *Schurger et al., 2012*)). This implies that having a clear movement goal does not necessarily mean that preparatory processes are fully engaged. Whether or not preparatory-subspace activity develops may depend on whether it is reasonably likely that movement will be initiated soon.

Our interpretation is that there exist at least three separable pre-movement processes: deciding what to do, preparing to do it, and initiating movement. It appears these processes occur with variable timing relative to one another. They occur very close in time (all within 100 ms) for the quasi-automatic context but are staggered for the self-initiated context. This interpretation is relevant to the finding that there exist pre-movement neural events that precede self-report of the decision to initiate movement (*Schurger et al., 2012*; *Libet et al., 1983*). In particular, preparatory-subspace activity develops hundreds of milliseconds before movement onset in the self-initiated context, presumably before a definitive choice to trigger movement has been made.

## Context-dependent movement generation

It seems a near-certainty that similar movements may be preceded and accompanied by divergent neural events depending on how and why a movement is prepared and initiated. As one example, while a prepared movement can be initiated in response to a learned visual cue, it can also be triggered involuntarily by the subcortically mediated startle reflex (*Valls-Solé et al., 1995*; *Carlsen et al., 2004*). As another example, the three contexts studied here evoke quite different movement-epoch responses in the supplementary motor area (*Lara et al., 2018*), despite the similarity of the movements themselves. Yet the present results reveal that motor cortex plays a more consistent role across contexts. While context impacted the timing of neural events, in the end, similar movements were driven by similar patterns of movement-related activity, preceded by similar patterns of preparatory activity.

# Materials and methods

## Subjects and task

Subjects were two adult male macaque monkeys (*Macaca mulatta*) aged 10 and 14 years and weighing 11–13 Kg at the time of the experiments. Daily fluid intake was regulated to maintain motivation to perform the task for juice reward. All procedures were in accord with the US National Institutes of Health guidelines and were approved by the Columbia University Institutional Animal Care and Use Committee (AC-AAAQ7409).

Subjects sat in a primate chair facing an LCD display and performed reaches with their right arm while their left arm was comfortably restrained. Hand position was monitored using an infrared optical system (Polaris; Northern Digital) to track (~0.3 mm precision) a reflective bead temporarily affixed to the third and fourth digits. Each trial began when the monkey touched and held a central touch-point. Touch-point color indicated context (*Figure 1*). After the touch-point was held for 450–550 ms (randomized) a colored 10 mm diameter disk (the target) appeared in one of eight possible locations radially arranged around the touch point. Target distance was 130 mm for cue and self-initiated contexts and 40 mm for the quasi-automatic context (*Figure 1*). Trials for different contexts/directions were interleaved using a block-randomized design.

In the cue-initiated context, after a variable delay period (0–1000 ms) the target suddenly grew in size to a diameter of 30 mm and the central touch point simultaneously disappeared. These events served as the go-cue, instructing the monkey to execute a reach. Reaches were successful if they were initiated within 500 ms of the go cue, had a duration <500 ms, and landed within an 18 mm radius window centered on the target. Juice was delivered if the monkey held the target, with minimal hand motion, for 200 ms (this criterion was shared across all three contexts).

In the self-initiated context, the target slowly and steadily grew in size, starting upon its appearance and ending when the reach began. Growth continued to a maximum size of 30 mm, which was achieved 1200 ms after target appearance (most reaches occurred before this time). The reward for a correct reach grew exponentially starting at 1 drop and achieved a maximum of 8 drops after 1200 ms. Monkeys were free to move as soon as the target appeared. However, monkeys essentially always waited longer in order to obtain larger rewards. Reward (and target size) ceased growing at the moment of movement onset. In rare instances where no movement was detected 1500 ms after target onset, the trial was aborted and flagged as an error. Requirements for reach duration and accuracy were as for the cue-initiated context.

In the quasi-automatic context, the target moved radially away from the central touch-point at 25 cm / s. Monkeys experienced both trials with and without a delay. In this study we consider only the latter. For these trials, motion began immediately upon target appearance. Target motion ended if a reach succeeded in bringing the hand to the target mid-flight. If the target was not intercepted then it continued moving until off the screen. Target speed and initial location (40 mm from the touch-point) were titrated, during training, such that the target was typically intercepted ~130 mm from the touch-point (the same location as the targets for the other two contexts). For successful interception, reaches had to land within an elliptical acceptance window (16 mm by 20 mm radius, with the long axis aligned with target motion). If the target was successfully intercepted, it grew in size to 30 mm and reward was delivered after the hold period.

This task was challenging, but monkeys successfully completed the majority of trials (>90% for every context and both monkeys). Percentages of different types of errors are given in *Table 1*. As documented, monkeys rarely 'jumped the gun' before target onset, even in the quasi-automatic context. A reasonable question is whether, if a delay had been included, monkeys would have been more likely to move during the delay for the quasi-automatic context. Although quasi-automatic trials with a delay are not considered here, monkeys did experience these trials (and were very familiar with them) and did occasionally make the mistake of reaching immediately after target onset, rather than respecting the delay.

## Visual display of stimuli

We used a photodetector (Thorlabs) to synchronize commands from the behavioral control software with the 60 Hz refresh schedule of the display. The photodetector monitored a white spot displayed in the corner of the screen that was masked from view by the monkey. Stimulus-display software (which ran in hard real time at 1000 Hz) waited for the spot to toggle (from off to on, or on to off) before sending commands for the next frame. Those commands included a command to toggle again. In this way, it was insured that all commands sent would register on the next frame, whose timing was known. Because the go cue was a particularly critical event, we used a second photodetector and spot to confirm its timing. The command for the go cue (e.g. the first frame when the target became enlarged for the cue-initiated context, or first moved for the quasi-automatic context) was accompanied by a command to turn on the second spot, and this event was recorded along with the behavioral data. This eliminates what would otherwise be a source of trial-to-trial variability in measurements of reaction time.

## Estimating movement onset

Movement onset was estimated based on hand speed: the magnitude of velocity. Optical tracking sampled hand position at 60 Hz. Because reaches contain negligible frequency content >15 Hz, appropriate filtering allowed hand velocity to be accurately estimated at every millisecond. Filtering was performed online and used to compute the time of movement onset for behavioral control purposes. The time of movement onset was re-measured offline (see below) after correcting for the 16 ms delay introduced by online filtering. A separate calibration confirmed that velocity-based

estimates of movement onset agreed with those obtained using a laser beam that became unbroken when the hand left the touch-point.

When estimating the time of movement onset (and thus RT) we wished to balance the constraints of accurately estimating the very first moment that the hand began to physically move, while minimizing variability in that estimate due to trial-to-trial noise or idiosyncrasies. That is, we wished to minimize bias while maintaining precision. Using a low speed threshold to estimate movement onset minimizes bias; the threshold is crossed very near the time the hand begins to move. However, a low threshold can introduce unwanted variability, due both to measurement noise and to slight trial-to-trial differences in the temporal profile of the very earliest changes in hand velocity. Such variability can be problematic when aligning trials to movement onset, because velocity profiles (and thus the activity of the neurons and muscles) can become misaligned. This can lead to high-frequency features of the response being smeared or distorted. We wished to minimize such misalignment without introducing a bias by using a higher threshold.

To do so, for each trial, we estimated the moment of peak acceleration: $t_{peakAcc}$. This identified an early feature of the response that occurred in a consistent fashion across trials; aligning trials on $t_{peakAcc}$ caused velocity profiles to be well aligned for their entire duration. We aligned all trials in that fashion, and computed the average speed. This was done separately for each condition. From the average speed profile, we found the time interval, $t_{corr}$, between when speed crossed a low threshold (1% of its peak) and the time of alignment, $t_{peakAcc}$. The RT for each trial was then $t_{peakAcc} - t_{corr}$. This method estimated well the moment when hand velocity first began to change (examples in *Figure 3*) while also producing good alignment of velocity profiles throughout the entire reach. The alternative method (estimating based on a low velocity threshold for each trial) occasionally resulted in poorly aligned velocity traces across trials, due to small differences in how hand velocity began for different trials. In rare instances a given trial had a velocity trace different from typical, such that it could not be well aligned with the other trials even using the preferred method. In such cases the trial was not analyzed.

## Neural and muscle recordings

After subjects became proficient in the task, we performed sterile surgery to implant a head restraint. At the same time, we implanted a recording chamber centered over the arm area of primary motor cortex (M1) and the dorsal premotor cortex (PMd. All recordings were from the left hemisphere and the task was performed with the right arm. Chamber positioning was guided by structural magnetic resonance images taken shortly before implantation. We used intra-cortical microstimulation to confirm that our recordings were from the forelimb region of motor cortex (biphasic pulses, cathodal leading, 250μS pulse width delivered at 333 Hz for a total duration of 50 ms). Microstimulation typically evoked contractions of the shoulder and upper-arm muscles, at currents from 5 μA – 60 μA depending on location and stimulation depth. We recorded single-neuron responses using traditional tungsten electrodes (FHC) or one or more 24-channel linear-array electrodes (V-probes; Plexon) lowered into cortex using a motorized microdrive. For tungsten-electrode recordings, spikes were sorted online using a window discriminator (Blackrock Microsystems). For linear-array recordings, spikes were sorted offline (Plexon Offline Sorter). We recorded all well-isolated task-responsive neurons; no attempt was made to screen for neuronal selectivity for reach direction or any other response property. Spikes were smoothed with a Gaussian kernel with standard deviation of 20 ms and averaged across trials to produce peri-stimulus time histograms.

We recorded electromyogram (EMG) activity using intramuscular electrodes from the following muscles: lower and upper aspects of the trapezius, medial, lateral and anterior heads of the deltoid, medial and lateral heads of the biceps, brachialis, pectoralis and latissimus dorsi. The triceps were minimally active and were not recorded. EMG signals were bandpass filtered (10–500 Hz), digitized at 1 kHz, rectified, smoothed with a Gaussian kernel with standard deviation of 20 ms, and averaged across trials to produce a continuous estimate of muscle activity intensity (very roughly analogous to a firing rate).

For analyses of latency (*Figure 3*, and the comparison between the onset of movement-subspace activity and EMG) we used a 10 ms Gaussian kernel, rather than 20 ms. We did so to minimize the small impact of filtering on the moment that activity first began to change. The method for assessing the onset of movement-subspace occupancy and EMG activity is described below.

## Data pre-processing prior to population analyses

As in our previous work, we employed two pre-processing steps (*Churchland et al., 2012*) before applying dimensionality reduction. First, the responses of each neuron were soft-normalized so that neurons with high firing rates had approximately unity firing-rate range (normalization factor = firing rate range + 5). This step ensures that subsequent dimensionality reduction (see below) captures the response of all neurons, rather than a handful of high firing-rate neurons. Second, the responses for each neuron were mean-centered at each time as follows: we calculated the mean activity across all conditions of each neuron at each time point, and subtracted this mean activity from each condition's response. This step ensures that dimensionality reduction focuses on dimensions where responses are selective across conditions, rather than dimensions where activity varies in a similar fashion across all conditions (*Kaufman et al., 2016*).

## Identifying preparatory and movement dimensions

We employed a recently developed dimensionality reduction approach. This approach leverages the finding that neural responses in the delay-epoch are nearly orthogonal to responses in the movement epoch (*Elsayed et al., 2016*). The method identifies one set of preparatory dimensions and an orthogonal set of movement dimensions. We defined two matrices based on data from the cue-initiated context only: $P \in R^{N \times CT}$ which holds preparatory epoch responses and $M \in R^{N \times CT}$ which holds movement-epoch neural responses. $N$ is the number of neurons recorded, $C$ is the number of reach directions and $T$ is the number of time points. The method seeks a set of preparatory dimensions, $W_{prep}$, that maximally capture the variance of $P$, and an orthogonal set of movement dimensions, $W_{move}$, that maximally capture the variance of $M$. We computed the preparatory and movement-epoch covariance matrices $C_{prep} = cov(P)$ and $C_{move} = cov(M)$ and optimized the following objective function:

$$\left[W_{prep}, W_{move}\right] = argmax_{[W_{prep}, W_{move}]} \frac{1}{2} \left( \frac{Tr\left(W_{prep}^T C_{prep} W_{prep}\right)}{\sum_{i=1}^{d_{prep}} \sigma_{prep}(i)} + \frac{Tr\left(W_{move}^T C_{move} W_{move}\right)}{\sum_{i=1}^{d_{move}} \sigma_{move}(i)} \right)$$

$$subject\ to\ \ W_{prep}^T W_{move} = 0, W_{prep}^T W_{prep} = I, \qquad W_{move}^T W_{move} = I$$

where $\sigma_{prep}(i)$ is the $i^{th}$ singular value of $C_{prep}$, and $\sigma_{move}(i)$ is the $i^{th}$ singular value of $C_{move}$. $W_{prep}$ and $W_{move}$ are the bases for the preparatory and movement subspaces respectively and $Tr(\cdot)$ is the matrix trace operator. $Tr\left(W_{prep}^T C_{prep} W_{prep}\right)$ is the preparatory-epoch data variance captured by the preparatory subspace, and $Tr\left(W_{move}^T C_{move} W_{move}\right)$ is the movement-epoch data variance captured by the movement subspace. We chose the dimensionality of $W_{prep}$ to be 12 (i.e. $W_{prep} \in R^{N \times 12}$), which captured ~80% of preparatory-epoch variance (the remaining variance had very little structure and appeared to be primarily sampling noise). Similarly, we chose the dimensionality of $W_{move}$ to be 12, which captured ~85% of movement-epoch variance. Results were robust with respect to the choice of dimensionality.

The optimization objective is normalized (by the singular values) to be insensitive to the relative dimensionality and amount of response variance in the two subspaces. This normalization is particularly important in the present application as movement activity is stronger and typically has higher dimensionality than the preparatory activity. Visualization required choosing two dimensions spanned by $W_{prep}$ and two by $W_{move}$ to define the plotted projections (e.g. in *Figure 8*). For $W_{prep}$ we chose the basis so that the top two dimensions captured the most variance (with all others ranked accordingly). For $W_{move}$ the basis was chosen using the jPCA method (*Churchland et al., 2012*) to capture movement-related oscillatory activity patterns. Visualization then employed the top two dimensions. The concatenation used to produce a unified estimate of firing rate versus time had essentially no impact on dimensionality reduction; the dimensions found are not sensitive to a discontinuity during the preparatory epoch (which was in any case very small for the cue-initiated context). What is critical is that the preparatory subspace was based entirely on data from well before

movement onset, and the movement subspace was found based entirely on data after muscle activity began.

## Projections and reconstructions

For a given time $t$ and for condition $\theta$, the projection of the population response onto the $k^{th}$ preparatory dimension is simply a weighted sum of all single-neuron responses: $x_k^{prep}(t,\theta) = \sum_{n=1}^{N} W_{n,k}^{prep} \, r_n(t,\theta)$ where $W_{n,k}^{prep}$ is the element in the $n^{th}$ row and $k^{th}$ column of $W_{prep}$ (see previous section) and $r_n(t,\theta)$ is the response of the $n^{th}$ neuron. This is illustrated in *Figure 6*, where the orange weights $w$ are taken from $W_{prep}$. The projection onto each movement dimension is defined analogously. The response of a given neuron can then be reconstructed as: $r_n(t,\theta) = \sum_{k=1}^{12} W_{n,k}^{prep} x_k^{prep}(t,\theta) + \sum_{k=1}^{12} W_{n,k}^{move} x_k^{move}(t,\theta)$. The two sums are the preparatory and movement contributions respectively, which we can term $r_n^{prep}(t,\theta)$ and $r_n^{move}(t,\theta)$. These are the preparatory and movement patterns for neuron $n$.

## Subspace occupancy

Our measurement of subspace occupancy is equivalent to the variance explained metric often computed in the context of principal component analysis. For the preparatory subspace, occupancy was computed as: $occupancy^{prep}(t) = \sum_{k=1}^{12} var_\theta\big(x_k^{prep}(t,\theta)\big)$ where $var_\theta$ indicates taking the variance across conditions (i.e. directions). Because dimensions are orthonormal this is equivalent to computing the variance for each neuron's preparatory pattern and then summing; that is, $\sum_{k=1}^{12} var_\theta\big(x_k^{prep}(t,\theta)\big) = \sum_{n=1}^{N} var_\theta\big(r_n^{prep}(t,\theta)\big)$. Movement subspace occupancy was defined analogously. To estimate the sampling error of the subspace occupancy we used a bootstrap procedure. We created 1000 surrogate populations by redrawing with replacement from the original population. We computed the subspace occupancy for each surrogate population. This yields, for each time, a sampling distribution of the occupancy. This sampling distribution was the basis of statistical measures. For example, the standard deviation of the sampling distribution yields the standard error of the mean, as plotted by the envelopes in *Figure 8*. P-values were also computed from this sampling distribution. For example, when comparing occupancy at a particular time across contexts, occupancy can be repeatedly redrawn from the two distributions. If 95% of such comparisons yield a value larger for one context versus the other, that corresponds to p=0.05 via a one-tailed test.

## Latency of physiological events

For some analyses we wished to compare timing of muscle activity with the timing of neural activity entering the movement subspace (last section of Results). For the neural data, we filtered the spike trains of all neurons using a Gaussian kernel with 10 ms standard deviation (rather than 20 ms for all other analyses of neural data). We recomputed the preparatory and movement dimensions using these data and calculated the subspace occupancy as before. We measured the latency as the first moment in time when occupancy reached 10% of peak occupancy.

Similarly, to calculate the latency of the EMG with respect to movement onset, we filtered EMG activity of all muscles using a Gaussian kernel with a 10 ms standard deviation. We then performed PCA on the EMG activity for each context separately and projected the corresponding EMG responses onto the first PC. We measured the latency as the first moment in which activity in the first PC reached 10% of peak activity. EMG was normalized prior to applying PCA, which is a critical step for ensuring that all muscles contribute to the PCs.

## Acknowledgments

We thank Yanina Pavlova and Sean Perkins for technical support. This work was supported by the Sloan Foundation, the Simons Foundation (SCGB#325171 and SCGB#325233), the Grossman Center for the Statistics of Mind, the McKnight Foundation, an NIH Directors New Innovator Award (DP2 NS083037), NIH/NSF CRCNS R01NS100066, NINDS U19 NS104649, P30 EY-019007, the Kavli

Foundation, a Klingenstein-Simons Fellowship, the Searle Scholars Program, an NIH Postdoctoral Fellowship (F32 NS092350) and the Gatsby Charitable Trust.

## Additional information

### Funding

| Funder | Grant reference number | Author |
|---|---|---|
| Alfred P. Sloan Foundation | | Mark M Churchland |
| Simons Foundation | SCGB#325171 | John P Cunningham<br>Mark M Churchland |
| Simons Foundation | SCGB#325233 | John P Cunningham<br>Mark M Churchland |
| Grossman Center for the Statistics of Mind | | John P Cunningham<br>Mark M Churchland |
| McKnight Endowment Fund for Neuroscience | | John P Cunningham<br>Mark M Churchland |
| National Institute of Neurological Disorders and Stroke | DP2 NS083037 | Mark M Churchland |
| National Institute of Neurological Disorders and Stroke | R01NS100066 | John P Cunningham<br>Mark M Churchland |
| National Institute of Neurological Disorders and Stroke | NS104649 | John P Cunningham<br>Mark M Churchland |
| National Eye Institute | P30 EY-019007 | Mark M Churchland |
| Kavli Foundation | | Mark M Churchland |
| Klingenstein Third Generation Foundation | | Mark M Churchland |
| Kinship Foundation | | Mark M Churchland |
| National Institute of Neurological Disorders and Stroke | NS092350 | Antonio H Lara |
| Gatsby Charitable Foundation | | John P Cunningham<br>Mark M Churchland |

The funders had no role in study design, data collection and interpretation, or the decision to submit the work for publication.

### Author contributions

Antonio H Lara, Conceptualization, Formal analysis, Investigation, Methodology, Writing—original draft, Writing—review and editing; Gamaleldin F Elsayed, Software, Formal analysis, Investigation, Methodology, Writing—review and editing; Andrew J Zimnik, Formal analysis, Validation, Writing—review and editing; John P Cunningham, Conceptualization, Supervision, Funding acquisition, Methodology, Project administration, Writing—review and editing; Mark M Churchland, Conceptualization, Formal analysis, Supervision, Funding acquisition, Validation, Writing—original draft, Project administration, Writing—review and editing

### Author ORCIDs

Mark M Churchland (iD) http://orcid.org/0000-0001-9123-6526

### Ethics

Animal experimentation: All procedures were in accord with the US National Institutes of Health guidelines and were approved by the Columbia University Institutional Animal Care and Use Committee (AC-AAAQ7409).

**Decision letter and Author response**
Decision letter https://doi.org/10.7554/eLife.31826.029
Author response https://doi.org/10.7554/eLife.31826.030

## Additional files

### Supplementary files

• Transparent reporting form
DOI: https://doi.org/10.7554/eLife.31826.024

### Data availability

The data supporting this work is available via Dryad (http://dx.doi.org/10.5061/dryad.cf66jb7).

The following dataset was generated:

| Author(s) | Year | Dataset title | Dataset URL | Database, license, and accessibility information |
| --- | --- | --- | --- | --- |
| Antonio Homero Lara, Gamaleldin F Elsayed, Andrew J Zimnik, John Cunningham, Mark M Churchland | 2018 | Data from: Conservation of preparatory neural events in monkey motor cortex regardless of how movement is initiated | http://dx.doi.org/10.5061/dryad.cf66jb7 | Available at Dryad Digital Repository under a CC0 Public Domain Dedication. |

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
