## [Decision Letter]

Thank you for submitting your article "Conservation of preparatory neural events regardless of how movement is initiated" for consideration by *eLife*. Your article has been reviewed by three peer reviewers, including Jennifer L Raymond as the Reviewing Editor and Reviewer #1, and the evaluation has been overseen by Timothy Behrens as the Senior Editor.

The reviewers have discussed the reviews with one another and the Reviewing Editor has drafted this letter to help you assess the concerns of the reviewers.

Summary:

This manuscript takes a systematic approach to investigating the nature of the preparatory activity, which is present in motor cortex before movement. This activity has been studied most extensively during motor tasks with an experimenter-imposed delay between the instruction and the "go" cue, to create a period during which the movement can be planned but not executed. In the current study, neural responses in motor cortex are compared during three reaching tasks: 1) cue-initiated, 2) self-initiated, and 3) a rapid, "quasi-automatic" reach to intercept a moving target. The authors build on elegant analytical approaches they introduced previously to decompose activity during the conventional, cue-initiated delay task into preparatory and movement-related subspaces. They then show that the structure of neural activity in these subspaces is preserved across conditions with greater urgency (quasi-automatic) or less urgency (self-initiated), though the time-course of events is modulated.

The experiments address an important outstanding question regarding whether or not attaining a certain preparatory state in motor cortex is a pre-requisite for generating a particular movement. The reviewers liked the general concept and analytical approach, however, they felt unconvinced by the conclusion that a particular preparatory state is required for movement, given the experimental logic and the data as currently presented.surname

Essential revisions:

1) The most critical problem is that the authors provide no convincing evidence that the monkeys are performing fundamentally different classes of movement, and the findings could be deemed quite trivial if one takes the view that the monkey has just been cued in different ways to generate the same movement. This is certainly true of the "self-initiated" condition. The very rapid condition is more interesting since in that case it has been argued that this represents a different class of movement perhaps not even relying on cortex. However, the reaction times (RTs) here are still rather long (~250 ms) relative to those found in the Perfiliev paper (~100-150ms). Thus, the reviewers were not convinced that the automatic task was eliciting automatic reaches, nor does it show "an almost innate short latency" consistent with Perfiliev et al., 2010, performed "near a physiological limit". The reaches are just generated slightly faster on some trials, and could still depend on preparation and so no wonder the neural activity is similar. Thus, in trying to match the kinematics across conditions, the authors might have failed to induce sufficient urgency in the "automatic condition". It would be easy to generate RT curves observed for the three tasks in the current study, with, say, the pre-cued situation but under different timing/task constraints. Consequently, this paper is limited to telling us that, even across a wide range of reaction times, some kind of cortical events are preserved, a new result perhaps but hardly a fundamental one, and certainly not the one claimed by the authors.

It is possible that the authors could address this issue with additional analysis. For example, it may be worthwhile to separately analyze trials in the "automatic" reach task with the very shortest reaction times. More generally, sorting trials within a given reach task by reaction time might provide additional evidence that the preparatory subspace is indeed capable of supporting preparation because it measures the state of preparation in a given context as revealed by the reaction time on a subset of trials.

2) Analysis of the robustness of the results to the analysis methods, and a measure of significance.

Rigor is lacking in the population analyses - the authors claim that no power analyses are known for the analyses they report. However, this is not the question that most-merits an answer. The issue is whether and how they know the reported results are significant. This requires a serious answer.

The dimensionality reduction approach and rationale are critical, but not really mentioned at all in the Results - only in the Materials and methods. Subsection “Segregating preparatory and movement responses at the population level”, first paragraph could easily be misconstrued as describing PCA done on separate epochs, which turned out to yield orthogonal subspaces (which was not the case). I would recommend being a little more specific here - i.e. explain that these are identified through a joint optimization process and constrained to be orthogonal to one another.

One issue which I suspect many readers will encounter is suspicion that the clean hand-off between preparatory and movement-related subspaces was an artifact of the dimensionality reduction algorithm. It would be reassuring in this regard to see that the "occupancy" of the null-space over time does not exhibit some systematic increase around the time of transition between preparation and initiation. If that did increase, this might suggest some interaction that is being overlooked by the dimensionality reduction. In particular, could there be some systematic activity in the self-initiated or quasi-automatic conditions that is not accounted for by activity in the cue-initiated condition?

"It is equally plausible that preparation has a sudden onset that is variable relative to movement onset, resulting in a ramp in the averaged data." This is an important point that currently barely gets mentioned. There seems no mention in the Discussion, despite abundant reference throughout to "ramping up" or "rising strength" of neural activity. It does not seem infeasible to distinguish in the current data between a gradual ramping up or a more abrupt transition that occurs at a variable time.

The first paragraph of the subsection “Single-neuron responses during the cue-initiated context” in the Results describes how the spike-trains taken from two distinct epochs were concatenated together, and then filtered to smooth out any discontinuity. It is unclear why this is necessary and this step is not mentioned at all in the methods. Is this stitching-together a critical pre-processing step applied to all the data prior to dimensionality reduction, or was it simply done to aid visualization? It is also unclear to what extent this stitching together of the data might have influenced the results and conclusions. I'm not too concerned that it would have, but I would expect the authors to more cautiously justify that it would not.

The authors sum up the behavior as mean peak velocity and a reaction time distribution, which is inappropriate given the key claim lies firmly within the details of this behavior. The authors also fail to discuss error trials. To me, this becomes especially notable in the context of the quasi-automatic case where this has been a parameter of great interest for the many papers that have looked at similar reflexive actions. Are the fastest reaction times in the quasi-automatic situations associated with more errors? If so, when does the animal actually start showing above chance directional performance?

[Editors’ note: this article was subsequently rejected after discussions between the reviewers, but the authors were invited to resubmit after an appeal against the decision.]

Thank you for submitting your work entitled "Conservation of preparatory neural events in monkey motor cortex regardless of how movement is initiated" for consideration by *eLife*. Your article has been reviewed by two peer reviewers, and the evaluation has been overseen by a Reviewing Editor and a Senior Editor. The reviewers have opted to remain anonymous.

Our decision has been reached after consultation between the reviewers. Based on these discussions and the individual reviews below, we regret to inform you that your work will not be considered further for publication in *eLife*.

The reviewers appreciated that the authors had made a serious effort to address the comments raised in the original review, and thought that the additional analyses and edits to the text had significantly improved the manuscript. Nevertheless, after discussion, the reviewers agreed that the lack of a reasonable number of sufficiently short reaction times severely limited the conclusions that could be made. The reviewers were just not convinced that the RTs for the quasi-automatic condition were fast enough to provide an interesting comparison with the cue-initiated condition.

Specific comments from the discussion among reviewers are included here, for the authors' information:

If there is such a thing as a distinct "automatic" movement mechanism, they may not have pushed the monkeys hard enough to engage in it. The average RTs of ~190ms are quite high compared to the 160ms in Perfiliev et al. (with still more or less perfect performance). Because the monkeys were not making any mistakes (error rate was very similar across all conditions), it is difficult to know whether they could have performed the task any faster, perhaps engaging an alternative mechanism if they did so. Had there been a clear decline in performance at very low RTs, they could have established a floor on the RT to base their low-RT analysis on and address this concern. As it is, all we can do is compare their monkey's behavior to RTs reported elsewhere in the literature.

The authors seek to get around the problem with the RTs being very different from Perfiliev by examining the latency of EMG responses. Target-specific EMG responses do seem to emerge within ~100ms, just as in Perfiliev et al., and the authors say this effect is driven by the very fastest trials. But there is a worrisome discrepancy between the EMG analysis and the RT analysis. Judging by Figure 2A, there seem to be very few trials with RTs below ~180ms. It certainly doesn't seem like enough trials to drive the early EMG effects in Figure 3, even given that EMG might precede movement by 30-50ms.

I don't think the median split on the RT is adequate. Since they are essentially claiming to be demonstrating the existence of preparatory activity during the Perfiliev effect, it is very important that they focus their analysis only on trials in which behavior is similar to that reported by Perfiliev et al., and the median split does not sufficient to ensure this. I worry that if they look at a lower percentile of RTs they won't have enough data left to establish whether or not there is preparatory activity.

The downward trend in preparatory occupancy for lower RTs is somewhat worrisome. One wonders how it might continue for even lower RTs.

Reviewer #2:

Overall, the authors have done a good job addressing the previous concerns and revising the manuscript. I believe the manuscript now provides an important and rigorous contribution to our understanding of movement generation by motor cortex.

I have just a couple of outstanding concerns:

Regarding the similarity of preparatory-subspace and movement-subspace activity across conditions: Establishing this similarity is the critical result in the paper (I don't think arguments based on occupancy alone are really sufficient to support the claim of 'preserved events'). To test whether movement-related activity is similar across contexts, the authors correlate activity across all 12 dimensions. However, to test whether planning-related activity is similar, the authors restrict their analysis only to the first two dimensions (i.e. the two with the greatest variance). The justification for this disparity feels a bit loose (that, for the preparatory subspace, the variance in the third dimension is only 14% that of the first dimension, while the variance is more evenly distributed across dimensions for the movement subspace). Why have a cut-off for the preparatory subspace, but no cut-off for the movement subspace? Given that the analysis in Figure 10 is based on covariance, shouldn't this naturally downweight dimensions in which there is not much variance? Are the results qualitatively similar if covariance is summed across all dimensions in this analysis?

Examining only short-RT trials in the quasi-automatic condition is important to support the central claim of the paper. However, the authors only extended this line of reasoning as far as the 'occupancy' analysis. Does the more fine-grained comparison of activity across contexts (i.e. Figs 9-10) hold when examining only short-RT quasi-automatic trials?

Reviewer #3:

In this revision the authors provide new analyses and significant revisions. The revisions improve the manuscript but do not really address the central underlying concern. The authors say we should not necessarily expect time-consuming preparation-like events to take place in an automatic task, but the events still take place. The problem is they don't state how long, or short, the time taken is that should be surprising, and they don't show that the automatic task offers an opportunity to perform the necessary test. We are still left with the concern that "the authors might have failed to induce sufficient urgency" in the automatic task which renders the results unsurprising.

The problem is that the automatic movements are generated with only slightly shorter RTs and not as quickly as the motivating study they cite. There is substantial overlap between the RTs associated with a delayed reach and an automatic reach. Despite the new data analysis, the revision does not assuage the original concern. In particular, the authors did not perform the analysis suggested, or they did not see a difference when they did. As a result, we still do not have a sense for how long RTs in the automatic task differ from short RTs in the delayed task, nor how the short RTs in the automatic task are particularly automatic. For those analyses provided, the results are not consistent with the expectation that the automatic reaches are indeed automatic. Instead the authors emphasize how the reach kinematics are the same for the two tasks, with almost no changes in endpoint accuracy, or evidence that the animals emit fast automatic reaches that then shift toward the target. This is claimed to be desirable because they want to control for the influence of kinematics on neural activity. However, this is a red-herring. The authors can subsample movements from the two different classes to have similar kinematics. What remains at issue from the first round of review is whether the movement categories differ in the first place.

---

## [Author Response]

[Editors’ note: the author responses to the first round of peer review follow.]

Essential revisions:1) The most critical problem is that the authors provide no convincing evidence that the monkeys are performing fundamentally different classes of movement, and the findings could be deemed quite trivial if one takes the view that the monkey has just been cued in different ways to generate the same movement. This is certainly true of the "self-initiated" condition. The very rapid condition is more interesting since in that case it has been argued that this represents a different class of movement perhaps not even relying on cortex. However, the reaction times (RTs) here are still rather long (~250 ms) relative to those found in the Perfiliev paper (~100-150ms). Thus, the reviewers were not convinced that the automatic task was eliciting automatic reaches, nor does it show "an almost innate short latency" consistent with Perfiliev et al., 2010, performed "near a physiological limit". The reaches are just generated slightly faster on some trials, and could still depend on preparation and so no wonder the neural activity is similar. Thus, in trying to match the kinematics across conditions, the authors might have failed to induce sufficient urgency in the "automatic condition". It would be easy to generate RT curves observed for the three tasks in the current study, with, say, the pre-cued situation but under different timing/task constraints. Consequently, this paper is limited to telling us that, even across a wide range of reaction times, some kind of cortical events are preserved, a new result perhaps but hardly a fundamental one, and certainly not the one claimed by the authors.It is possible that the authors could address this issue with additional analysis. For example, it may be worthwhile to separately analyze trials in the "automatic" reach task with the very shortest reaction times. More generally, sorting trials within a given reach task by reaction time might provide additional evidence that the preparatory subspace is indeed capable of supporting preparation because it measures the state of preparation in a given context as revealed by the reaction time on a subset of trials.

The reviewers express a straightforward and sensible concern: the clear presence of preparation across contexts might not be deeply surprising if those contexts are not terribly different in the first place. The reviewers raise the concern that the ‘very rapid condition’ (the quasi-automatic context) might not be that rapid compared with the original Perfiliev study. The reviewers suggest that this concern could be addressed with an additional analysis: separately analyzing trials with particularly short RTs. We understand the nature of these concerns and have made multiple changes to address them. Most importantly this includes performing the requested analysis, and adding a new analysis showing that EMG response latencies in our study are in fact nearly identical to those in Perfiliev. We have also heavily revised aspects of the paper including the Introduction. We agree that these additions were needed and are glad that the reviewers motivated us to make them. These additions and changes are enumerated below.

i) We have performed the requested analysis. The motivation for that request is as follows: if our response latencies are on average longer than in Perfiliev, then that shortcoming could be mitigated if we down-selected our data to approximate Perfiliev’s range of latencies. As will be addressed below, our response latencies, based on EMG, are in fact comparable to those in Perfiliev (this just wasn’t clear in the original manuscript). That said, we still very much like the suggested analysis: it provides an additional test of the hypothesis that a preparatory stage occurs even for particularly short-latency movements. The results of this new analysis are shown in the new Figure 8 and are clear: the preparatory subspace is occupied for both short-RT trials and long-RT trials. In fact, in both cases, peak preparatory subspace occupancy was actually slightly higher for the quasi-automatic context relative to the cue-initiated context. This is important because it had historically been controversial whether delay-period-like events occurred during zero-delay trials, even of the ‘normal’ sort. The new analysis showi.e.i.e.s that not only do they occur in the absence of a delay, they do so even for a context that evokes low-latency responses, and even when analyzing particularly low-latency trials. Text describing this new analysis begins in the subsection “The preparatory subspace is occupied for short-RT quasi-automatic trials”.)

ii) We have added a new figure documenting that the quasi-automatic context produces EMG response latencies nearly identical to those in Perfiliev et al. The new Figure 3 includes examples of raw EMG on individual trials, and an analysis of average EMG with minimal filtering (both requested by the reviewers). This new analysis demonstrates that, when making an apples-to-apples comparison between our responses and those of Perfiliev, the latency of muscle activity is virtually identical. Perfiliev et al. reported that “the earliest onset of electromyographic activity… ranged from 90 to 110 ms”. Thus, in their study, the trials with the shortest latencies exhibited EMG that began changing 90-110 ms (depending on the subject) following the onset of target motion. Our findings replicate theirs almost perfectly: the earliest onset of EMG occurred ~100 ms after target onset. We apologize that this was not documented before, and are glad to have been motivated to make this important addition.

iii) The Introduction, and sections of the Results, have been revised to highlight a key reason why we employed the self-initiated context: it distinguishes between the competing hypotheses that delay period activity is preparatory versus suppressive. These hypotheses produce opposing predictions in the self-initiated context. Given the existing controversy on this topic (see below), the self-initiated context provides a very useful test. We apologize that this was not better explained in the original submission.

iv) The Introduction has been heavily written to convey why, given the literature, it was controversial whether delay-period-like activity would appear without a delay. The reviewers note that it was unclear from the original manuscript how surprised a reader should be that putatively preparatory activity (i.e., activity in the same subspace and with the same structure as delay-period activity) reappears in other contexts. We understand the nature of this concern. If you accept that delay period activity is preparatory, then that predicts that similar activity will likely reappear if there is time to prepare. Yet critically, that prediction is premised on a specific hypothesis: that delay period activity is fundamentally preparatory and consistently occurs, in a similar way, before movements made under very different amounts of time pressure. That hypothesis has been controversial for a variety of reasons. The Introduction has been restructured to convey these points.

v) The Introduction has been rewritten to highlight that the quasi-automatic context is intrinsically a zero-delay condition, and that the existing literature has yielded ambiguous results regarding whether putatively preparatory activity appears for zero-delay trials. In our original submission, we failed to adequately stress that multiple prior studies (including the recent influential Ames et al., 2014 study) have yielded ambiguous results regarding whether delay-period events reappear during zero-delay conditions (even ‘normal’ zero-delay trials without moving targets).The Introduction now contains the following passage: “We also compared neural activity with and without a delay, but employed a different analysis approach and compared neural activity across a broader range of behavioral ‘contexts’... To examine preparation without a delay, we did not simply reduce the cue-initiated delay to zero (as in Crammond and Kalaska, 2000; Churchland et al., 2006; Ames, Ryu and Shenoy, 2014), but instead adopted the paradigm of Perfiliev et al., 2010 and used moving targets... This ‘quasi-automatic’ context provides a particularly stringent test of the prediction that delay-period-like activity will reappear in the absence of a delay.” This new text highlights that we are revisiting the ‘classic’ comparison between trials with and without a delay, with the additional benefit of using zero-delay trials with very short RTs. This provides a particularly stern test of whether preparation occurs.

vi) We have made a small correction to our velocity-based RT measurements, allowing for a more appropriate comparison with prior literature. Although EMG-based response latencies in our study were nearly identical to those in Perfiliev et al., we observed a slightly longer delay between EMG onset and detectible changes in hand velocity. Thus, as the reviewers noted, our velocity-based RTs are slightly longer than theirs. This may relate to the fact that reaches had to break the friction of the screen in our task (mechanical factors are known to impact the time from EMG onset to movement onset; e.g., Anson et al., 1982, J Mot. Beh.). However, 16 ms of discrepancy occurs for a trivial reason that we should have corrected for.Perfiliev et al. used a 500 Hz optical tracking system to identify the onset of hand velocity. We used a 60 Hz tracker, and online filtering converted the resulting ‘staircase’ into a 1000 Hz stream used for behavioral control. Because hand movement has negligible frequency content above 15 Hz, filtering produces an essentially perfect estimate of hand position at each millisecond. However, that estimate follows the ‘trailing’ edge of the staircase rather than the leading edge; i.e. it occurs exactly 16 ms late relative to what would be measured by a 1000 Hz tracking system.We should have corrected for this lag in our original calculations and have now done so. The correction doesn’t change comparisons internal to the study (the RT difference between cue-initiated and quasi-automatic contexts remains identical) but corrects a small bias and thus improves the ability to compare with other studies including the important Perfiliev study. We have also now performed a calibration procedure using a laser beam and a photodetector. This verified the accuracy of the velocity-based RT measurements. These details, along with other procedures to ensure accurate RT measurements are now described in the Results (subsection “Reaction times (RTs)”). The new Figure 3 shows examples of when movement was detected, on top of single-trial velocity and EMG traces.

2) Analysis of the robustness of the results to the analysis methods, and a measure of significance.

We understand and agree with these criticisms, and have made multiple additions to address them. These are detailed in turn below.

Rigor is lacking in the population analyses - the authors claim that no power analyses are known for the analyses they report. However, this is not the question that most-merits an answer. The issue is whether and how they know the reported results are significant. This requires a serious answer.

We have added p-values for all key observations. We agree that in the original manuscript we relied overly heavily on standard error bars, which provide only an indirect measure of significance. We have now run significance tests based on the same bootstrap procedure that produced the error bars. The new Table 2 gives p-values for multiple key comparisons. In other cases, p-values and/or 95% confidence intervals are given in the text. For example, in multiple cases we use correlation to compare events within across contexts. The text gives 95% confidence intervals and/or p-values for these comparisons.

The dimensionality reduction approach and rationale are critical, but not really mentioned at all in the Results - only in the Materials and methods. Subsection “Segregating preparatory and movement responses at the population level”, first paragraph could easily be misconstrued as describing PCA done on separate epochs, which turned out to yield orthogonal subspaces (which was not the case). I would recommend being a little more specific here - i.e. explain that these are identified through a joint optimization process and constrained to be orthogonal to one another.

We agree. The following paragraph has been added to the Results:

“The simplest method for identifying putatively preparatory and movement-related dimensions is to apply principal component analysis separately to delay-epoch and movement-epoch data. […] This achieves a result similar to principal component analysis, but ensures all dimensions are truly orthogonal.”

One issue which I suspect many readers will encounter is suspicion that the clean hand-off between preparatory and movement-related subspaces was an artifact of the dimensionality reduction algorithm. It would be reassuring in this regard to see that the "occupancy" of the null-space over time does not exhibit some systematic increase around the time of transition between preparation and initiation. If that did increase, this might suggest some interaction that is being overlooked by the dimensionality reduction. In particular, could there be some systematic activity in the self-initiated or quasi-automatic conditions that is not accounted for by activity in the cue-initiated condition?

The reviewers raise a number of good questions. We have added a section to the Results, ‘Methodology and the segregation between preparatory and movement subspaces’.This section addresses the above issues, including how results change (typically very little) if we identify subspaces using data from the self-initiated context rather than the cue-initiated context. We had actually already performed this analysis as a sanity check, but had not described it in the original submission.

"It is equally plausible that preparation has a sudden onset that is variable relative to movement onset, resulting in a ramp in the averaged data." This is an important point that currently barely gets mentioned. There seems no mention in the Discussion, despite abundant reference throughout to "ramping up" or "rising strength" of neural activity. It does not seem infeasible to distinguish in the current data between a gradual ramping up or a more abrupt transition that occurs at a variable time.

We agree that this is an interesting (and timely) line of inquiry and have added a new analysis on this topic (Figure 7—figure supplement 1). We had previously avoided this line of inquiry for two reasons. First, although important, it was somewhat separate from the main questions of our study. Second, we didn’t think we would be able to address this topic without very large-scale simultaneous recordings. However, motivated by the reviewers, we realized there was a simple analysis that we could perform. This analysis addresses whether activity is a ramp of variable slope, or instead follows a stereotyped temporal profile beginning a variable time after target onset. The new analysis does not attempt to resolve the exact profile of preparatory activity on single trials, but nevertheless yields some compelling results. We are glad to have been motivated to add it. This new analysis is described in the Results:

“Rising preparatory-subspace occupancy in the self-initiated context could reflect steady ramping on individual trials (with short RTs associated with a steeper ramp) or a more sudden onset at a variable time. […] Whether that development is a stereotyped ramp (lasting a few hundred milliseconds), or a step with variable timing on individual trials (with a distribution spanning a few hundred milliseconds), cannot be readily determined from the present data.”

The first paragraph of the subsection “Single-neuron responses during the cue-initiated context” in the Results describes how the spike-trains taken from two distinct epochs were concatenated together, and then filtered to smooth out any discontinuity. It is unclear why this is necessary and this step is not mentioned at all in the methods. Is this stitching-together a critical pre-processing step applied to all the data prior to dimensionality reduction, or was it simply done to aid visualization? It is also unclear to what extent this stitching together of the data might have influenced the results and conclusions. I'm not too concerned that it would have, but I would expect the authors to more cautiously justify that it would not.

The reviewers are correct on both counts: we did not adequately describe our method for concatenation and it does not impact any of the reported results. We have added markers to Figures 4, 7 and 8, noting the points of concatenation. We have also added a section to the Results describing how concatenation was performed:

“For visualization purposes, we wished a continuous estimate of firing-rate spanning target-driven and movement-related events. […] Doing so produces a representative trace and minimizes the small discontinuity in firing rate at the time of concatenation.”

We also added text to the Materials and methods noting that dimensionality reduction is insensitive to the (very small) discontinuity produced by concatenation:

“The concatenation used to produce a unified estimate of firing rate versus time had essentially no impact on dimensionality reduction; the dimensions found are not sensitive to a discontinuity during the preparatory epoch (which was in any case very small for the cue-initiated context). What is critical is that the preparatory subspace was based entirely on data from well before movement onset, and the movement subspace was found based entirely on data after muscle activity began.”

The authors sum up the behavior as mean peak velocity and a reaction time distribution, which is inappropriate given the key claim lies firmly within the details of this behavior. The authors also fail to discuss error trials. To me, this becomes especially notable in the context of the quasi-automatic case where this has been a parameter of great interest for the many papers that have looked at similar reflexive actions. Are the fastest reaction times in the quasi-automatic situations associated with more errors? If so, when does the animal actually start showing above chance directional performance?

We are pleased to be motivated to add further analyses documenting their behavior – our monkeys performed the task both consistently and we are quite happy to add further documentation of their behavior!

As requested, we have added a table (Table 1) documenting the percentage of every type of error.

Regarding the reviewer’s question, “Are the fastest reaction times in the quasi-automatic situation associated with more errors?”, the manuscript contains the following (revised) section addressing this question:

“Is some price in accuracy paid for such rapid responses? [...] In summary, quasi-automatic reaches were slightly less accurate, but were otherwise similar to reaches in the other contexts.”

Regarding the related question, “when does the animal actually start showing above chance directional performance?”, we have added a new analysis of initial reach-angle distributions for the quasi-automatic context, with distributions computed separately for short-RT and long-RT trials (Figure 2—figure supplement 2). This analysis demonstrates that reaches are essentially always target directed; reaches are never directed randomly as if the animal were guessing. This analysis is described in the Results:

“Might monkeys sometimes begin quasi-automatic reaches before they are fully specified? (Ghez et al., 1997). […] Finally, the early EMG response was target-specific for most muscles across all contexts (Figure 3—figure supplement 1).”

To illustrate that average trajectories are representative, we have added a new supplementary figure (Figure 2—figure supplement 1) showing individual-trial trajectories for both monkeys and all three contexts.

With these additions, the revised manuscript now contains analyses of all key aspects of behavior, including:

i) RT distributions (Figure 2A, B) and median RTs (text) for all contexts and both monkeys.

ii) Average reach trajectories for all targets / contexts and both monkeys (Figure 2C, D).

iii) Individual-trial reach trajectories for all targets/contexts and both monkeys (Figure 2—figure supplement 1).

iv) Average velocity profiles for all targets/contexts and both monkey (Figure 2E, F).

v) Analysis of the component of movement orthogonal to the screen (Figure 2G, H).

vi) Distributions of initial reach angles for the quasi-automatic context for short-RT versus long-RT trials (Figure 2—figure supplement 2).

vii) Comparison of relative peak speed for quasi-automatic versus cue-initiated movements (subsection “Reach kinematics”).

viii) Comparison of reach angle variability for quasi-automatic versus cue-initiated movements (subsection “Reach kinematics”).

ix) Comparison of target miss-rate for quasi-automatic versus cue-initiated movements (subsection “Reach kinematics”).

x) Examples illustrating the relationship between the onset of EMG and the rise in hand velocity on single trials (Figure 3).

xi) Percentages of different error modes for all contexts and both monkeys (Table 1).

xii) Percentages of successful trials for all contexts and both monkeys (Results, fourth paragraph) [As a minor technical note, in the revised submission item xii was recomputed using a more conservative criterion regarding when a trial was initiated (versus never attempted in the first place). Because a trial had to be initiated to be subsequently failed, this resulted in a very small change (0-2%) in the numbers reported versus those in the original manuscript. We note this simply to make clear that there is no unaccounted-for discrepancy.]

[Editors’ note: the author responses to the re-review follow.]

Our study was rejected for one straightforward reason: a belief that, for the quasi-automatic context, responses were not as short-latency as in Perfiliev et al. This is summarized at the top of the review:

“the lack of a reasonable number of sufficiently short reaction times severely limited the conclusions that could be made.”

This claim is incorrect. Our responses are not longer-latency than in Perfiliev. If anything, our responses are slightly shorter latency.

We believed the low-latency of responses would be adequately conveyed by Figure 3, which showed that trial-averaged EMG begins changing ~100 ms after target onset, consistent with the 90-110 ms earliest onset of EMG in Perfiliev. That point was further illustrated via example trials with latencies of 100-150 ms (the average in Perfiliev was 144 ms). However, Figure 3 received the following criticisms:

i) Perhaps only a small handful of our trials have early EMG onset latencies. If so, plots of average EMG could be misleading.

ii) Perhaps the example trials we chose are unusually fast and un-representative.

In fact, both average EMG plots and the examples are quite representative. We have added new figures (Figure 4 and Figure 4—figure supplements 1 and 2) to reinforce this. These figures document EMG latencies on single trials. This is done for muscles with low baseline activity, such that the onset of EMG can be detected on single trials. These figures clarify that EMG latencies are as at least as short as in Perfiliev.

In addition to adding these figures to the manuscript, we also include below tables and Author response image 1. These make the following points:

**Author response image 1. respfig1:** Comparison of EMG between humans and monkeys. All data are for the quasi-automatic context. MC and AZ are two of the authors. Yellow traces show average target-locked EMG for all eight target directions. Latency is estimated manually as the first moment when activity begins to separate for the different directions. Recordings are from the *deltoid*. We chose the *deltoid* as the most natural point of comparison because the shoulder was the primary joint responsible for producing the reach. The *deltoid* had a ‘smooth’ onset: initially changing slowly and then more rapidly. Other muscles (e.g., the *biceps*) had a more abrupt onset. Average target-locked hand velocity is shown by gray-to-black shaded traces (one per target direction). One might have expected EMG onset to be later in humans given the greater length of cortico-spinal axons. However, conduction velocities in the corticospinal tract are very fast (exceeding 100 m/s).

i) Our average EMG latencies are shorter, not longer, than in Perfiliev (117-142 ms in our task versus 144 ms in theirs).

ii) Our earliest EMG latencies are essentially identical to those in Perfiliev (85-109 ms in our task versus 90-110 ms in theirs).

iii) The median split yields average latencies considerably shorter than the average latency in Perfiliev (102-121 ms versus 144 ms). Thus, analysis performed after the median split is concentrated on very short-latency trials.

iv) Human performance on our task produces EMG latencies and EMG-to-velocity lags essentially identical to those in monkey. In particular, the earliest EMG onset is 90-110 ms, just as for our monkeys, and just as in Perfiliev.

Inspection of these figures and tables should eliminate any concern that there is a lack of sufficiently short responses in our task.

Following the documentation below, a standard point-by-point reply is appended. That reply describes useful extensions to existing analyses, as requested by reviewer #2. These have been added to the manuscript.

Mean EMG latencies:

Perfiliev

humans144 ms SD across subjects = 20 ms, estimated range of 119-169 ms

Numbers from p. 2426 of Perfiliev et al. The SD describes across-subject variability. Range is estimated as +/- 1.25 * SD, and thus contains ~80% of the individual-subject means. The range is thus conservative: a few subjects are expected to lie outside this range.

Lara

monkey Ba118 ms *trapezius*
117 ms *biceps*monkey Ax142 ms *biceps*

EMG latency is based on single-trial measurements for muscles where baseline activity was low, allowing EMG onset to be unambiguously identified (see Figure 4 and Figure 4—figure supplements 1 and 2).

Lara, trials with latencies <= median

monkey Ba102 ms *trapezius*
103 ms *biceps*monkey Ax121 ms *biceps*

At the reviewers’ request, we had added analyses examining preparatory subspace activity after restricting to trials faster than the median. We note that further restriction would have minimal impact on the mean analyzed RT. For example, the numbers above are reduced only slightly (by 7.9 ms, 4.6 ms and 6.4 ms) if restricted to the fastest quartile.

Earliest EMG latencies:

Perfiliev

humans90-110 ms

Earliest onset was measured for each subject. The range is across subjects. See p. 2426 of Perfiliev.

Lara

monkey Ba85 ms *trapezius*
89 ms *biceps*monkey Ax109 ms *biceps*

Based on single-trial measurements as above.

Lara, based on target-locked average EMG

monkey Ba90 ms *multiple muscles, including deltoid*monkey Ax110 ms *multiple muscles, including deltoid*human AZ90 ms *deltoid*human MMC110 ms *deltoid*

Estimated via inspection of when target-locked averages first began to change. This method provides a good way of estimating the earliest latency for muscles with baseline activity (where single-trial estimates are not reliable). Estimates agree with those made via single trials but are 1-5 ms more conservative.

Human results are not included in our manuscript. Human data were collected only for this rebuttal, and demonstrate that the earliest EMG latencies, and the EMG-to-velocity lag, are similar for humans and monkeys. See below.

Is there a discrepancy between our EMG latencies and hand-velocity-based latencies?

The review notes that ‘Judging by Figure 2A, there seem to be very few trials with RTs below ~180 ms’ which ‘doesn't seem like enough trials to drive the early EMG effects in Figure 3, even given that EMG might precede movement by 30-50ms.’ As stated in the text, RTs for our two monkeys were 205 ± 32 ms and 192 ms ± 27 ms (mean and SD after pooling across all sessions). Given these SDs, one expects RTs < 180 ms and this was indeed the case: 21% (Ba) and 36% of trials had RTs < 180 ms. Furthermore, 12% and 21% had RTs < 170 ms. EMG onset precedes movement onset by 40-80 ms (depending on the muscle and monkey, see below) not ‘30-50 ms’. Thus, EMG latencies as early as 90-110 ms are entirely expected.

Thus, the numbers are correct. However, the reviewers were correct in noting something slightly wrong with the corresponding figure: all traces were shifted a few pixels to the right and are thus not centered on the numbers given in the text. This was a small error, but of course hurt the point we were trying to make. We are not sure how this mistake occurred. The figure generated by matlab was/is correct. But at some point, presumably while panels were assembled in Illustrator, a mistake was introduced. This has been rectified. We have also had both the senior author and one of the middle authors re-compute mean RTs. All measurements agree within one millisecond.

Is there a discrepancy between our hand-velocity-based latencies and those of Perfiliev?

The EMG-to-velocity lag in our task fully explains why our velocity-based RTs are slightly longer than in Perfiliev, even though our EMG latencies tended to be slightly *shorter*. We observed lags of ~40-80 ms, depending on the muscle and monkey. These lags are an expected consequence of both physiology and our task geometry: the hand rests on a screen whose considerable friction helps the monkey to keep his hand stationary. It is difficult for hand velocity to build until the hand is lifted slightly off the screen. Had our screen been pressure sensitive, we would presumably have found slightly shorter physical latencies.

Given the empirical 40-80 ms lags, our velocity-based RTs *should* be at the long end of the range in Perfiliev. The magnitude of this effect is exactly as expected, and is not as large as stated in the review. In the review, velocity-based RTs from Perfiliev are described (at different points) as being ‘100-150 ms’ and ‘160 ms’. The 100-150 range is incorrect. 160 ms is a better estimate, but still doesn’t give credit to the considerable subject-to-subject variability in Perfiliev:

Average velocity-based RTs, Perfiliev

**Monkeys**
120-200 ms Range is across subjects. Group mean not given.Humans, physical target139-189 ms Group mean = 164 ms, across-subject SD = 20 ms.Humans, computer monitor148-194 ms Group mean = 171 ms, across-subject SD = 18 ms.Cats160-200 ms Range is across subjects. Group mean not given.

For humans, ranges are not given in Perfiliev. We estimated them as 1.25 times the across-subject SD. This ensures the range encompasses ~80% of subjects. Ranges are thus conservative: a few subjects will exceed this range.

Average velocity-based RTs, Lara

monkey Ba205 msmonkey Ax192 ms

Thus, our velocity-based RTs have an entirely sensible relationship with those in Perfiliev. For example, monkey Ba had a mean *biceps* response latency of 117 ms (*faster* than the 144 ms mean in Perfiliev). The mean *biceps*-to-velocity lag was 79 ms, yielding a mean RT of 196 ms on that session (at the long end of the Perfiliev range for monkeys).

How do human and monkey EMG latencies compare?

When we perform the task ourselves, we produce very similar EMG latencies and similar EMG-to-velocity lags.

The reviewers appreciated that the authors had made a serious effort to address the comments raised in the original review, and thought that the additional analyses and edits to the text had significantly improved the manuscript.Nevertheless, after discussion, the reviewers agreed that the lack of a reasonable number of sufficiently short reaction times severely limited the conclusions that could be made. The reviewers were just not convinced that the RTs for the quasi-automatic condition were fast enough to provide an interesting comparison with the cue-initiated condition.

As discussed above, there is not in fact a lack of ‘a reasonable number of sufficiently short reaction times’. Indeed, our latencies are generally slightly faster than in Perfiliev.

Nearly all the major concerns were related to this fundamental issue. Thus, addressing this issue addresses many of the concerns below. (Reviewer 2 also makes helpful requests for extensions to analyses, which we have performed).

Specific comments from the discussion among reviewers are included here, for the authors' information:If there is such a thing as a distinct "automatic" movement mechanism, they may not have pushed the monkeys hard enough to engage in it. The average RTs of ~190ms are quite high compared to the 160ms in Perfiliev et al. (with still more or less perfect performance). Because the monkeys were not making any mistakes (error rate was very similar across all conditions), it is difficult to know whether they could have performed the task any faster, perhaps engaging an alternative mechanism if they did so. Had there been a clear decline in performance at very low RTs, they could have established a floor on the RT to base their low-RT analysis on and address this concern. As it is, all we can do is compare their monkey's behavior to RTs reported elsewhere in the literature.

This concern is largely addressed by the documentation above, showing that our EMG latencies are at least as fast as in Perfiliev, and that our velocity-based RTs are only slightly different (and for an expected reason).

Monkeys do miss the target more than twice as often for the quasi-automatic context versus the cue-initiated context, as stated in the text. Another failure mode is revealed for a trial-type not analyzed in the present study: quasi-automatic trials with a delay. Although very practiced at these, monkeys still had an increased tendency to reach immediately following target onset and fail the trial. This is now stated in the Materials and methods:

“As documented, monkeys rarely ‘jumped the gun’ before target onset, even in the quasi-automatic context. […] Although quasi-automatic trials with a delay are not considered here, monkeys did experience these trials (and were very familiar with them) and did indeed often make the mistake of reaching immediately after target onset, rather than respecting the delay.”

We do not believe or argue that there is a distinct automatic movement mechanism. Nowhere in our manuscript do we suggest this – indeed we caution against it.

The authors seek to get around the problem with the RTs being very different from Perfiliev by examining the latency of EMG responses. Target-specific EMG responses do seem to emerge within ~100ms, just as in Perfiliev et al., and the authors say this effect is driven by the very fastest trials. But there is a worrisome discrepancy between the EMG analysis and the RT analysis. Judging by Figure 2A, there seem to be very few trials with RTs below ~180ms. It certainly doesn't seem like enough trials to drive the early EMG effects in Figure 3, even given that EMG might precede movement by 30-50ms.

This is addressed by the rebuttal above.

I don't think the median split on the RT is adequate. Since they are essentially claiming to be demonstrating the existence of preparatory activity during the Perfiliev effect, it is very important that they focus their analysis only on trials in which behavior is similar to that reported by Perfiliev et al., and the median split does not sufficient to ensure this. I worry that if they look at a lower percentile of RTs they won't have enough data left to establish whether or not there is preparatory activity.

The extensive documentation above demonstrates that the median split is more than adequate. Even without the split, our latencies are already slightly faster on average than in Perfiliev. After the median split, our latencies are quite a bit faster (see the third table under heading ‘Mean EMG latencies’ above).

Further restricting analysis would result in only a small (~10 ms, see above) further shortening of mean latency. And as the review notes, there would then be too few trials to analyze. For example, if we were to analyze only the fastest quartile, some neurons would contribute only 1-2 trials per condition. This would mean that we would be essentially performing a single-trial analysis, which our method is not meant to handle (it lacks an appropriate noise model). Even the median-split was a bit of a stretch (we could count on only >= 3 trials per condition). Empirically it worked well and yielded results that were not dominated by noise. Yet we would not wish to push this further (and essentially nothing would be gained by trying to do so).

The downward trend in preparatory occupancy for lower RTs is somewhat worrisome. One wonders how it might continue for even lower RTs.

As noted above, RTs really can’t get much lower. Trials faster than the median are already very fast and further down-selection would have minimal further effect.

More critically, there isn’t really a ‘downward trend’. The critical question of our study is whether events that occur during an imposed delay also occur in the absence of that delay. This was indeed the case: preparatory-subspace occupancy was always *at least as high* for quasi-automatic trials (with no delay) as for the corresponding cue-initiated trials (with a delay). We have simplified Figure 9 to better highlight this central finding. Importantly, yellow traces (quasi-automatic context) *always peak higher* than red traces (cue-initiated context). Thus, there is no evidence that delay-period-like events are absent or reduced in the quasi-automatic context – if anything they are slightly stronger.

A *separate* effect is that peak preparatory-subspace occupancy is slightly higher for long-RT trials, and slightly lower for short-RT trials. This goes against the common (but incorrect) notion that higher firing rates should be associated with more rapid responses. Indeed, these lower peaks are expected: the event that prompts movement onset creates a rise in occupancy. Occupancy is then truncated just before movement onset. Later onsets thus give more time for the rise to occur before it is truncated. This effect can be observed in the new Figure 10—figure supplement 2, where the pattern of states for short-RT trials expands in the same way as for all trials, but then start to contract earlier. However, we stress that this is a small effect – it seems sizeable in Figure 9 only because occupancy is a squared measure. This is now better highlighted in the text:

“We also observed that, across all comparisons, long-RT trials tended to have slightly higher preparatory-subspace peaks compared to short-RT trials. […] Subsequent state-space analyses show this effect more directly, and also reveal that it is very modest (it is magnified when considering occupancy, which quantifies variance).”

We have also added Figure 11—figure supplement 1. This figure confirms that, for short-RT trials, the neural states in the preparatory subspace achieve a similar arrangement across contexts. Data are slightly noisier after the down-selection, but there is certainly no lack of preparatory subspace activity for short-RT trials.

Reviewer #2:Overall, the authors have done a good job addressing the previous concerns and revising the manuscript. I believe the manuscript now provides an important and rigorous contribution to our understanding of movement generation by motor cortex.I have just a couple of outstanding concerns:Regarding the similarity of preparatory-subspace and movement-subspace activity across conditions: Establishing this similarity is the critical result in the paper (I don't think arguments based on occupancy alone are really sufficient to support the claim of 'preserved events'). To test whether movement-related activity is similar across contexts, the authors correlate activity across all 12 dimensions. However, to test whether planning-related activity is similar, the authors restrict their analysis only to the first two dimensions (i.e. the two with the greatest variance). The justification for this disparity feels a bit loose (that, for the preparatory subspace, the variance in the third dimension is only 14% that of the first dimension, while the variance is more evenly distributed across dimensions for the movement subspace). Why have a cut-off for the preparatory subspace, but no cut-off for the movement subspace? Given that the analysis in Figure 10 is based on covariance, shouldn't this naturally downweight dimensions in which there is not much variance? Are the results qualitatively similar if covariance is summed across all dimensions in this analysis?

We have extended the correlation-based analysis of preparatory states to all 12 dimensions. These results support our original conclusions. We originally avoided using all dimensions because, past the first two, they mostly contribute noise. This choice also allowed quantification to link directly to the 2D plots. However, we agree that it seems odd to use a cutoff for the preparatory subspace but not the movement subspace. The reviewer is also correct that covariance/correlation-based analyses naturally down-weight low-variance dimensions. Results for 12 dimensions are thus very similar to results for 2 dimensions:

correlations are slightly reduced due to added noise but are still high (r = 0.73-0.95). This is now described:

“Results were similar if we considered all twelve preparatory dimensions. Although higher dimensions primarily captured noise, their inclusion reduced the correlation only moderately given that they contributed little variance. […] Correlations were intermediate (0.77-0.98) when measured in six dimensions.”

The covariance-based analysis in Figure 11C, D (which had been Figure 10C, D) was always based on all 12 dimensions (though as the reviewer anticipated, the number of dimensions analyzed makes essentially no difference to this analysis – results are virtually identical either way). The fact that we used 12 dimensions was admittedly not at all clear from the manuscript. It is now stated in the figure legend.

Examining only short-RT trials in the quasi-automatic condition is important to support the central claim of the paper. However, the authors only extended this line of reasoning as far as the 'occupancy' analysis. Does the more fine-grained comparison of activity across contexts (i.e. Figs 9-10) hold when examining only short-RT quasi-automatic trials?

[Note that Figures 9 and 10 are now Figures 10 and 11]

These results do indeed hold when examining for the short-RT quasi-automatic trials. We have included a new figure (Figure 11—figure supplement 1) repeating the key analysis (Figure 11A, B) for short-RT trials. Results are slightly noisier (presumably due to the reduced trial count) but still quite clear.

We also added Figure 10—figure supplement 2 comparing the evolution in state space for all quasi-automatic trials versus short-RT quasi-automatic trials. This figure allows one to observe that, for short-RT trials, the same events occur but unfold slightly earlier, due to the earlier movement onset.

Reviewer #3:In this revision the authors provide new analyses and significant revisions. The revisions improve the manuscript but do not really address the central underlying concern. The authors say we should not necessarily expect time-consuming preparation-like events to take place in an automatic task, but the events still take place. The problem is they don't state how long, or short, the time taken is that should be surprising, and they don't show that the automatic task offers an opportunity to perform the necessary test. We are still left with the concern that "the authors might have failed to induce sufficient urgency" in the automatic task which renders the results unsurprising.

The premise of this concern is that the presence of delay-period-like events without a delay is interesting (‘surprising’) *only* if latencies are shorter than some difficult-to-determine threshold. It is indeed true that the presence of delay-period-like events is particularly compelling when responses are short latency. The documentation above underscores that we have achieved very short latencies. However, we would still like to largely disagree with the premise. Our study is not just (or even primarily) about examining the neural basis of ‘the Perfiliev effect’. As described in the Abstract and Introduction, our study is about testing the hypothesis that the events that occur during a delay deserve to be called preparatory. A critical test of that hypothesis is whether delay period-like events appear in the absence of a delay. This has been controversial, and recent studies (e.g., Ames et al.) have been taken as evidence that delay-period events are not a ‘normal’ part of the sequence of movement generation. We are frequently asked at meetings (including as recently as a few weeks ago) whether we believe that delay-period activity would ‘naturally’ be seen without a delay, or whether it results from the artificial situation of an experimenter-imposed delay. This is a fundamental question, and multiple studies (Ames et al., Crammond and Kalaska, our own prior work) have tried to address it by comparing long-delay trials with plain-old garden-variety zero-delay trials.

In the present study we are in a better position to address this fundamental question because we are able to segregate putatively preparatory and movement-related events via subspace. In leveraging this new ability, it would have been quite acceptable (as in prior studies) to simply compare long-delay trials with zero-delay trials (i.e., to use only the cue-initiated context). That said, we certainly agree that the quasi-automatic context provides a particularly stringent test of whether delay-period-like events occur without a delay, and we used it for that reason. We have revised the text to further highlight this motivation:

“The low latency of the quasi-automatic reaches aids interpretation when asking whether delay-period-like events occur without a delay. […] Rather, we use the quasi-automatic context as a stringent test of the hypothesis that delay-period events are preparatory and will thus typically occur – if only briefly – even in the absence of a delay.”

The problem is that the automatic movements are generated with only slightly shorter RTs and not as quickly as the motivating study they cite.

This is addressed in the extensive rebuttal above. Our latencies are at least as fast as in Perfiliev.

There is substantial overlap between the RTs associated with a delayed reach and an automatic reach.

We do not agree that this is a problem – indeed it would be concerning if there weren’t some overlap. For cue-initiated trials, the monkey benefits from a delay and can prepare in advance. If RTs for the cue-initiated context had a mean RT so long that it eliminated overlap, that would imply that the monkeys are likely not taking advantage of the delay and are not preparing in advance. We are thus pleased that RTs in the cue-initiated context are relatively brisk. This is stated in the text:

“Cue-initiated RTs were 253 ± 50 ms and 235 ± 37 ms (mean ± s.d.) for monkey Ba and Ax (Figure 2A, B,red traces). These RTs are on the brisk side of the range in prior studies, consistent with the goal of the cue-initiated context: to encourage monkeys to prepare during the delay and reach promptly after the go cue.”

Despite the new data analysis, the revision does not assuage the original concern. In particular, the authors did not perform the analysis suggested, or they did not see a difference when they did.

We performed the analysis exactly as outlined in the ‘action plan’ submitted for approval. The median split is more than sufficient: mean EMG onset latencies for short-RT trials are noticeably faster than the mean Perfiliev latency (see “Mean EMG latencies” section above). Further splitting the data into smaller and smaller groups would yield diminishing returns (e.g., only an ~10 ms speedup if we restricted to the fastest quartile). Furthermore, we would then be left with so little data (1-2 trials per condition for some neurons) that we would not trust the results. We would be close to performing a single-trial analysis. We 100% stand by our decision to use the median split, and to not attempt analyses that would yield little additional value and would likely become difficult to interpret. It is certainly *not* the case that we performed a highly restricted analysis but failed to report results we did not like.

As a result, we still do not have a sense for how long RTs in the automatic task differ from short RTs in the delayed task, nor how the short RTs in the automatic task are particularly automatic. For those analyses provided, the results are not consistent with the expectation that the automatic reaches are indeed automatic. Instead the authors emphasize how the reach kinematics are the same for the two tasks, with almost no changes in endpoint accuracy, or evidence that the animals emit fast automatic reaches that then shift toward the target. This is claimed to be desirable because they want to control for the influence of kinematics on neural activity. However, this is a red-herring. The authors can subsample movements from the two different classes to have similar kinematics. What remains at issue from the first round of review is whether the movement categories differ in the first place.

Our motivations for using the quasi-automatic context are not based on any presumption of different categories of movement. The Abstract and Introduction explain our motivations (also see above) and make no reference to different categories. Indeed, we work to dispel the idea that we believe the different contexts are fundamentally different categories. The original submission contained the following:

“However, we stress that this term should not be taken to imply that quasi-automatic reaches are necessarily a different class of movement at the level of motor cortex.”

The revised submission contained the following:

“This allows us to revisit the traditional comparison between trials with a long delay versus no delay (Crammond and Kalaska, 2000; Churchland et al., 2006; Ames, Ryu and Shenoy, 2014). The quasi-automatic context provides a stringent version of this comparison; the moving target evokes low-latency responses (see below) raising the question of whether preparation is even possible. The term ‘quasi-automatic’ should not be taken to imply a different class of movement at the neural level. It is possible that quasi-automatic reaches involve a dedicated mechanism for rapid initiation. Yet quasi-automatic reaches could also represent an extreme version of ‘zero-delay’ trials, with high time-pressure and a particularly effective go cue. In either case, a key question is whether movement onset is preceded by putatively preparatory events.”

We have further revised the text:

“The low latency of the quasi-automatic reaches aids interpretation when asking whether delay-period-like events occur without a delay. This question has traditionally been addressed by including zero-delay trials that are otherwise identical to long-delay trials^6,11,22^. While valid, that approach raises a potential concern: might monkeys, once accustomed to a delay, start to treat zero-delay trials as if they have a delay? For example, monkeys might increase their RT to accommodate the expectation of a delay, and might begin to generate delay-period-like neural events that would otherwise not occur. The low-latency of quasi-automatic trials addresses such concerns. Monkeys cannot be ‘adding’ their own delay, otherwise RTs would be longer rather than shorter than for the cue-initiated context. Indeed, quasi-automatic responses are sufficiently swift that it becomes reasonable to question whether preparation is possible^20,23^, given that preparation is usually proposed to be time-consuming.[…] That said, we do not propose that the quasi-automatic trials represent a different ‘class’ of movement. Rather, we use the quasi-automatic context as a stringent test of the hypothesis that delay-period events are preparatory and will thus typically occur – if only briefly – even in the absence of a delay.”